The body plan of Halszkaraptor escuilliei (Dinosauria, Theropoda) is not a transitional form along the evolution of dromaeosaurid hypercarnivory

Cau Andrea cauand@gmail.com
Independent , Parma , Italy
Knoll Fabien
Electronic publication date: 2020 Feb 25
Publication date: 2020
Volume: 8
Electronic Location ID: e8672
Received 2019 Dec 13; Accepted 2020 Jan 30
Copyright: © 2020 Cau
Copyright year: 2020
Copyright holder: Cau
License: This is an open access article distributed under the terms of the Creative Commons Attribution License, which permits unrestricted use, distribution, reproduction and adaptation in any medium and for any purpose provided that it is properly attributed. For attribution, the original author(s), title, publication source (PeerJ) and either DOI or URL of the article must be cited.
License URL: https://creativecommons.org/licenses/by/4.0/

Keywords: Theropoda, Maniraptora, Cretaceous, Mongolia, Dinosauria, Phylogenetics, Halszkaraptorinae, Dromaeosauridae, Homoplasy

Funding: The author received no funding for this work.

==============================
The dromaeosaurid theropod Halszkaraptor escuilliei is characterized by several unusual features absent in other paravians, part of which has been interpreted as diagnostic of a novel lineage adapted to a semiaquatic ecology. Recently, these evolutionary and ecological interpretations have been challenged, and Halszkaraptor has been claimed to be a transitional form between non-dromaeosaurid maniraptoriforms and other dromaeosaurids: following that reevaluation, its peculiar body plan would represent the retention of several maniraptoran plesiomorphies, lost among other dromaeosaurids, and not an adaptation to a novel ecology. This alternative scenario is here carefully investigated and tested. It is shown that most statements supporting this scenario are based on misinterpretation of anatomical traits and bibliography. Once these statements have been corrected, character state transition optimization over a well-supported phylogenetic framework indicates that the large majority of the peculiar features of the Halszkaraptor lineage are derived novelties acquired by the latter after its divergence from the last ancestor shared with eudromaeosaurs, and thus are not maniraptoriform plesiomorphies. At least seven novelties of the Halszkaraptor lineage are convergently acquired with spinosaurids, and are integrated in semiaquatic adaptations: one of these is reported here for the first time. The amount of morphological divergence of Halszkaraptorinae from the ancestral dromaeosaurid condition is comparable to those of Microraptorinae and Velociraptorinae. Among extant taxa, the sawbills (Mergini, Anseriformes) show the closest ecomorphological similarity with the peculiar body plan inferred for Halszkaraptor. The halszkaraptorine bauplan is thus confirmed as a derived amphibious specialization, and does not represent a “transitional” stage along the evolution of dromaeosaurids.

Introduction

The bird-like theropod dinosaur Halszkaraptor escuilliei is a based on an almost complete skeleton from the Upper Cretaceous of Mongolia (Cau et al., 2017). Compared to other theropods, Halszkaraptor shows several unusual features, supporting the institution of a new lineage of Dromaeosauridae, the halszkaraptorines (Cau et al., 2017; Cau & Madzia, 2018; Agnolin et al., 2018), and suggesting a semiaquatic bauplan able to exploit both terrestrial and aquatic resources. Recently, Brownstein (2019) published a review of the interpretations of Cau et al. (2017), and concluded that Halszkaraptor was not a semiaquatic form but a “transitional form” between the plesiomorphic maniraptoriform bauplan and the hypercarnivorous dromaeosaurids. Here, I show that several statements in Brownstein (2019) are unsupported, inaccurate or contradictory, and that most of the arguments raised by Brownstein (2019) stem from a substantial misinterpretation of the literature (in primis, but not uniquely, Cau et al., 2017) or are based on problematic homology statements.

Materials and Methods

Brownstein (2019) cited several statements from the literature in support of his arguments: they were carefully checked and when not corresponding to the original source, they were reported and commented. All information from H. escuilliei discussed here was acquired by AC at the Royal Belgian Institute of Natural Sciences (RBINS), Brussels, where MPC D-102/109 is temporarily housed (Cau et al., 2017). First-hand examination of MPC D-102/109 was integrated with multi-resolution scan data of the fossil, based on propagation X-ray phase-contrast synchrotron microtomography performed in 2016, at the European Synchrotron Radiation Facility in Grenoble, France (Cau et al., 2017). In order to test the evolutionary and phylogenetic scenario suggested by Brownstein (2019), I used a new version of the data set used in Cau et al. (2017) (Supplemental Files). The data matrix was analyzed using TNT vers. 1.5 (Goloboff, Farris & Nixon, 2008), following the same protocol of Cau et al. (2017: a first round of 100 “New Technology” runs, using default setting, was followed by a Tree-Bisection-Reconnection run using the shortest trees saved during the first round as starting topologies). The Triassic dinosaur Herrerasaurus was used as root of the trees. Four spinosaurid taxa were included in the sample, to test the distribution of the features shared by Halszkaraptor and those non-coelurosaurian theropods (Cau et al., 2017). The “agreement subtree” algorithm implemented in TNT was used to reconstruct the taxonomically most comprehensive fully-dichotomous structure shared by all shortest trees found: for this reconstruction, max tree was set to 50.000 due to memory limitations in TNT. The agreement subtree topology was used as framework for character state reconstruction at nodes and for estimating the minimum length of the recovered branches. Character state transition reconstruction at nodes was performed in PAUP (Swofford, 2002), importing the agreement subtree topology reconstructed in TNT and using the ACCelerated TRANsformation (ACCTRAN) optimization. Taxonomic nomenclature follows Cau et al. (2017), with emendation of Unenlagiinae following Hartman et al. (2019). The distribution of the reconstructed state transitions along the theropod phylogeny was used to compare the alternative scenarios discussed by Cau et al. (2017) and Brownstein (2019).

Results

Several sentences in Brownstein (2019) are inaccurate or problematic, including mentions to statements in the literature which are actually contradicted by the mentioned references themselves. In the following references, the term “Ref./Refs.” followed by one or more numbers refers to the reference list in Brownstein (2019).

Literature misreports and unsupported statements

Brownstein (2019) compared the premaxilla of Halszkaraptor with those of ornithomimosaurs and therizinosaurians. He wrote: “moderately to strongly (=platyrostral) laterally expanded premaxillae are found in a variety of maniraptorans and maniraptoriforms […]. Among these, the premaxillae of Erlikosaurus are the best preserved and are highly reminiscent of the premaxillae of Halszkaraptor in their clear lateral expansion in dorsal view […]”. Contra Brownstein (2019), it is unlikely that the platyrostral morphology of Halszkaraptor is homologous to those he referred to other coelurosaurs. In Halszkaraptor, the platyrostral condition is acquired by the remarkable anteroposterior elongation and dorsoventral flattening of the prenarial region of the premaxilla, which also results in the posterior placement of the narial region relative to the snout anterior tip. In ornithomimosaurs, the platyrostral condition is instead related to the lateral expansion of the perinarial region (Osmólska, Roniewicz & Barsbold, 1972; Lee et al., 2014) which is not followed by any significant elongation of the prenarial region. In Erlikosaurus, the prenarial part of the premaxilla is taller than long, constrasting with the opposite condition in Halszkaraptor (Fig. 1). Note that the relative elongation of the prenarial part of the premaxilla and the posterior retraction of the premaxillary margin of the external naris are not co-variant and thus could be considered as independent features (e.g., Haplocheirus, Choiniere et al., 2014b). In Erlikosaurus, the narial fossa is expanded laterally and forms the majority of the premaxillary body, whereas in H. escuilliei the narial fossa is completely excluded from the participation to the premaxillary body (Fig. 1). The “lateral expansion” of the premaxilla, claimed by Brownstein (2019), is thus produced by distinct elements in the two taxa (i.e., prenarial elongation and depression in Halszkaraptor, vs sub- and perinarial widening in Erlikosaurus), and could not be considered homologous (Fig. 1).

Figure 1 Comparison between the skull of the therizinosaurid Erlikosaurus andrewsi (MPC D-100/111: A and C) and the paravian Halszkaraptor escuilliei (MPC D-102/109: B and D), in left lateral (A and B) and dorsal (C and D) views.

Key differences in snout morphology: prenarial part of premaxilla taller than long (a1) or longer than tall (a2); platyrostral condition produced by perinarial widening (b1) or prenarial flattening (b2); complete loss of premaxillary dentition (c1) or supranumerary premaxillary dentition (c2); maxillary dentition lacking replacement waves (d1), or bearing distinct replacement waves (d2); narial fossa widely overlapping premaxillary oral margin (e1) or narial fossa not overlapping premaxillary oral margin. Scale bars in mm. (A) and (C) Provided by Stephan Lautenschlager (used with permission).

Brownstein (2019) wrote: “It is unclear how Cau et al. [Ref. 32] observed retracted nares in Halszkaraptor, as the anterior nasals are not preserved in that taxon”. Based on his own statement, Brownstein (2019) assumed that the retraction of the external naris in Cau et al. (2017) was meant as the position of the narial margin of the nasal. As clearly stated in the latter paper, the retraction of the external naris referred to the narial margin of the premaxilla, and not to the narial margin of the nasal. They wrote: “[T]he platyrostral premaxilla with a dorsolaterally oriented external naris that is, retracted beyond the oral margin is unique among theropods, although in its elongation, the premaxilla is similar to those of spinosaurids” (Cau et al., 2017). Brownstein (2019) thus raised a concern for a feature which actually was not discussed by Cau et al. (2017). Furthermore, Brownstein (2019) wrote: “Despite the support for it found here, if the presence of elongate nares is not found as the plesiomorphic state for coelurosaurs in future analyses, the presence of them in a variety of theropods that do not show any features for a semiaquatic lifestyle provides evidence against the argument of Cau et al. [Ref. 32], who argued this feature was indicative of such an ecology”. Contra Brownstein’s (2019) claim, Cau et al. (2017) did not argue that the “elongation” of the naris is present in Halszkaraptor or that it is relevant in whatever ecological scenario. The actual feature mentioned by Cau et al. (2017), the posterior retraction of the premaxillary narial margin beyond the premaxillary body, is absent in alvarezsauroids (Choiniere et al., 2014b), ornithomimosaurs (Osmólska, Roniewicz & Barsbold, 1972; Lee et al., 2014), oviraptorosaurs (Balanoff et al., 2009; Balanoff & Norell, 2012) and therizinosauroids (Lautenschlager et al., 2014), and is instead comparable to that in baryonychine spinosaurids (Charig & Milner, 1997; Sereno et al., 1998), where it has been interpreted as a piscivorous adaptation (Charig & Milner, 1997; Milner, 2003; Rayfield et al., 2007; see discussion in Hone & Holtz (2017)).

Brownstein (2019) wrote: “Although Halszkaraptor was differentiated from other theropods in possessing a rostral neurovascular system not entirely restricted [to] the lateral portions of the premaxillae [Ref. 32], the rostral neurovasculature extends onto the dorsal surface of the body of the premaxilla in basal members of most other maniraptoran clades”. Several statements by Brownstein (2019) inaccurately listed the external distribution and the density of the neurovascular foramina in other theropods. Contra Brownstein (2019), the premaxilla of Shenzhousaurus is much less extensively pitted than in Halszkaraptor (see Ji et al., 2003, [Ref. 13] cited by Brownstein (2019)). Contra Brownstein (2019), the neurovascular foramina in ornithomimosaurus are densely distributed only along the oral margin but are less extensively distributed (if not absent) along the rest of the premaxillary body (Kobayashi & Lü, 2003; Ksepka & Norell, 2004; Kobayashi & Barsbold, 2005; Lee et al., 2014). The same condition is present in oviraptorosaurs (Balanoff et al., 2009; Balanoff & Norell, 2012), where the premaxilla is extensively pitted only along the oral margin and scarcely penetrated in the rest of the bone. In all mentioned examples, the condition in these taxa differs from Halszkaraptor, where the density of foramina is greater and their distribution is more extensive all along the bone surface. Among theropods, only spinosaurids show a comparable density and distribution of neurovascular foramina in the premaxilla (Charig & Milner, 1997; Dal Sasso et al., 2005; Ibrahim et al., 2014, Fig. S6).

Furthermore, Brownstein (2019) wrote: “In the more derived therizinosaur Erlikosaurus, the same morphology, where the premaxillae harbor neurovascular foramina on both their lateral and mediodorsal surfaces, is clearly present (see Lautenschlager et al. [Ref. 19] for clear scans of the premaxillae of Erlikosaurus; Figs. 1C and 1D)”. Contra Brownstein (2019), the CT-scanning of Erlikosaurus (Lautenschlager et al., 2014) demonstrates that both density and number of the external foramina and the relative size of the internal plexus in that therizinosaurid are much less developed than in Halszkaraptor (Fig. 2). In Erlikosaurus, the external foramina are mainly concentrated along the oral margin (Lautenschlager et al., 2014), and are less numerous in both absolute and relative terms than in Halszkaraptor (in the latter, the dorsal surface bears at least 20 foramina). Given that neurovascular foramina number is expected to positively correlate with premaxilla size, the higher number of foramina in Halszkaraptor (which bears a premaxilla about 1/5 the size of that of Erlikosaurus) is thus a very unusual condition. Furthermore, the relative size of the internal neurovascular plexus (including its main stem) in Halszkaraptor is significantly larger than in Erlikosaurus (Fig. 2).

Figure 2 Development of the premaxillary neurovascular plexus in some archosaurs.

Semitransparent rendering of premaxilla of Erlikosaurus andrewsi (MPC D-100/111: A and B) in lateral (A) and dorsal (B) views. Semitransparent rendering of premaxillae of Halszkaraptor escuilliei ( MPC D-109/109: C and D) in lateral (C ) and dorsal (D) views. Semitransparent rendering of anterior end of snout in Crocodylus sp. (uncatalogued specimen: E), Halszkaraptor escuilliei (F) and Erlikosaurus andrewsi (MPC D-100/111: G) in dorsal view. Semitransparent rendering of snout in cf. Spinosaurus aegyptiacus (MSNM V4047: H and I) in lateral (H) and dorsal (I) views. (A–D) and (E–I) Rescaled at same width for comparison. In red, rendering of the neurovascular plexus. Arrows in E–I indicate the level of the anterior margin of the external naris. (A), (B) and (G) Modified from images provided by Stephan Lautenschlager (used with permission). (H) and (I) Modified from images provided by Dawid Adam Iurino (used with permission). Abbreviations: en, external naris; nps, basal stem of the neurovascular plexus; pnr, prenarial part of premaxilla.

Brownstein (2019) focused his discussion on the tooth replacement patterns in Halszkaraptor and assumed that its condition was comparable to those of omnivorous maniraptoriforms. He wrote: “One interesting feature of the premaxillary teeth of Halszkaraptor described by Cau et al. [Ref. 32] was their delayed replacement rate. A large amount of research into the loss of teeth in some maniraptoran dinosaurs has found a delayed replacement rate to be linked to tooth loss in several clades, including therizinosaurs and ornithomimosaurs” (italics added here), and “the slowly-replacing premaxillary teeth of Halszkaraptor are also reminiscent of adaptations found in herbivorous theropod lineages like therizinosaurs [Refs. 11 and 15]”. The above mentioned sentences misinterpreted and misreported the literature. First, Cau et al. (2017) described a “delayed replacement pattern” in Halszkaraptor, and not a “delayed replacement rate”. The term “delayed replacement pattern”, used by Cau et al. (2017) for Halszkaraptor, refers to the differences between the premaxillary tooth-replacement compared to the maxillary tooth-replacement. Second, Refs. 11 and 15 mentioned by Brownstein (2019), that is, Zanno et al. (2009), and Zanno & Makovicky (2011), do not mention “delayed replacement rate” but instead “low tooth-replacement rate”. The “low tooth-replacement rate” described by Zanno et al. (2009) and Zanno & Makovicky (2011) refers to the absence of pronounced replacement waves and gaps between teeth, producing a continuous horizontal cutting surface (L. Zanno, 2019, personal communication). Brownstein (2019) thus misinterpreted the terminology used by Cau et al. (2017) and assumed that Zanno et al. (2009) and Zanno & Makovicky (2011) referred to the same condition described by Cau et al. (2017). Contra Brownstein (2019), the two terms are relative to distinct, non-homologous conditions, and are not synonyms. The “low tooth-replacement rate” (Zanno et al., 2009; Zanno & Makovicky, 2011) is absent in Halszkaraptor, which bears a sinusoid cutting surface along the whole dentition and distinct replacement waves (see below, Fig. 3). Brownstein (2019) also stated that the “delayed replacement rate is linked to tooth loss”. That sentence is not correct: the absence of replacement waves in the teeth is independent to the loss of teeth, as demonstrated by Pelecanimimus (Perez-Moreno et al., 1994), parvicursorines (Chiappe, Norell & Clark, 1998) and several troodontids (Lü et al., 2010), all lacking replacement waves yet retaining a complete set of teeth (Zanno & Makovicky, 2011). Given that Halszkaraptor is unique among all dinosaurs in having the largest number of premaxillary teeth (Cau et al., 2017), it shows a condition opposite to the loss of premaxillary teeth seen in therizinosaurids or other omnivorous/herbivorous theropods (Zanno & Makovicky, 2011). In sum, Halszkaraptor is not “reminiscent of adaptations found in herbivorous theropod lineages” (contra Brownstein, 2019).

Figure 3 Premaxillae and maxillae of H. escuilliei MPC D-102/109 in right lateral view.

In (A), the different bones are colored to help the identification of the distinct elements forming the rostrum. Note that the majority of the right maxilla is lost (light blue), revealing most of the left maxilla (pink) in medial view (in his Fig. 1, Brownstein (2019), misinterpreted the preservation of the maxillae and depicted most of the lateral surface of the right maxilla based on the medial side of the left one). In (B), semi-transparent reconstruction of the same elements, showing the tooth roots and the “festooning” pattern in tooth size variation. Scale bar in mm. Abbreviations: lmx, left maxilla; lpmx, left premaxilla; m1-2, first and second maxillary tooth; pdl, paradental lamina; rmx, right maxilla; rpmx, right premaxilla.

Brownstein (2019) wrote: “Cau et al. [Ref. 32] noted the comparatively long neck of Halszkaraptor […]. However, […], it is unclear why Cau et al. allied this feature to elongate necks in derived semiaquatic avians (e.g., Cygnus)”. Cau et al. (2017) did not compare Halszkaraptor neck elongation to the condition in derived semiaquatic avians (e.g., Cygnus). In the latter study, Cygnus is only mentioned once, but in relation to the shape of the interpostzygapophyseal lamina and not because of its neck elongation. Brownstein (2019) misunderstood two distinct sentences in Cau et al. (2017) and combined them improperly. Brownstein (2019) wrote: “Despite the fact that Cau et al. [Ref. 32] claimed the neck of Halszkaraptor composed the greatest percentage of snout-to-sacrum length among non-avian coelurosaurs, a large number of clades include taxa that approach, reach, or possibly even exceed that threshold” (Italics added here). The above statement misreports the original sentence of Cau et al. (2017), which instead was: “[c]ompared to body size, the neck is elongate and forms 50% of the snout–sacrum length; this is the highest value found among Mesozoic paravians thus far” (italics added here).

Brownstein (2019) wrote: “If this hypothesized ecomorphology for Halszkaraptor is correct, it has major implications for the evolution of bird-like dinosaurs, with H. escuilliei representing the first aquatic non-avian maniraptoran and suggesting that the ancestral lifestyle for dromaeosaurids could be one that took place in the water [Ref. 32]”. Contra Brownstein (2019), Cau et al. (2017) [his Ref. 32] did not suggest that the ancestral dromaeosaurid lifestyle could be one that took place in the water.

Brownstein (2019) wrote: “The Djadokhta Formation […] preserves a highly arid environment […]. Given this environmental setting, it is hard to envision that specialized, semiaquatic dromaeosaurs would populate this ecosystem”. Other Djadokhtan reptiles show adaptations related to an amphibious lifestyle, like the neosuchian Shamosuchus diadochtaensis (a taxon characterized by a platyrostral snout and unserrated subconical dentition; Pol, Turner & Norell, 2009). Even if not the most abundant members, semiaquatic taxa are present in the Djadokhtan faunal assemblages (Lefeld, 1971): their relatively low frequency is in agreement with the presence of ephemeral lacustrine deposits in that Formation (Dingus et al., 2008), but does not constitute a challenge to the ecological interpretation of Cau et al. (2017).

Misreports and misinterpretation of Halszkaraptor anatomy

Brownstein (2019) wrote: “In many dromaeosaurids, including velociraptorines, Halszkaraptor, Deinonychus and “Bambiraptor”, the anterior end of the ventral surface of the dentary bulges to form a chin, as in some ornithomimosaurs”. Brownstein (2019) included a drawing of the anterior end of the dentary of Halszkaraptor in lateral view, and depicted a distinctly convex “bulge” at the anterior end of the ventral margin (indicated in that figure by an arrow). That drawing is inaccurate and misleading. The anteroventral end of the dentary in Halszkaraptor is eroded (Figs. 1B and 1D), so its exact shape, including the presence of the “bulge” illustrated by Brownstein (2019), cannot be determined. Regardless to what actually Brownstein (2019) meant with “bulge” in the dromaeosaurid dentaries, claiming its presence in Halszkaraptor is not based on evidence.

Brownstein (2019) wrote: “On the whole, the skull of Halszkaraptor also shares many similarities with basal troodontids, including […] tightly packed teeth, and recurved, ziphodont, unserrated crowns”. The combination of terms “ziphodont” coupled with “unserrated” is a contradiction due to the improper use of the anatomical terminology: “ziphodont” means, literally, tooth with serration (Langston, 1975). The dentition of Halszkaraptor is not ziphodont. In Halszkaraptor, only the premaxillary teeth are packed, a condition comparable to Microraptor but absent in other microraptorines (Xing et al., 2013; Pei et al., 2014). The rest of the dentition (notably, the whole maxilla) in H. escuilliei is formed by spaced alveoli with complete interdental septa, differing from troodontids where the anterior maxillary dentition is formed by tightly packed teeth housed in a sulcus often lacking interdental septa (Lü et al., 2010). The topographical differences between the regions bearing packed teeth suggest that the condition in H. escuilliei is not homologous to that in troodontids.

Brownstein (2019) wrote: “Shortened caudal series. Halszkaraptor possesses a highly modified caudal series, a feature that Cau et al. [Ref. 32] used to support a modified posture in this taxon analogous to some birds”. Cau et al. (2017) did not state that the caudal series of Halszkaraptor is shortened. The actual number of caudal vertebrae in MPC D-102/109 is unknown, being the distal end of the tail missing. The preserved part of the tail in MPC D-102/109 is comparable to the majority of paravians (Godefroit et al., 2013b; Lefèvre et al., 2017) in elongation and proportions of the vertebrae, and it is not significantly reduced, as instead seen in pygostilian birds or in some oviraptorosaurs (Zhou et al., 2000; Cau, 2018). Furthermore, Cau et al. (2017) did not write that the unusual features in the caudal vertebrae of Halszkaraptor support a modified posture like that in birds: the latter was inferred on the basis of hypertrophied origin and insertion of the m. ileofibularis in, respectively, ilium and femur (Cau et al., 2017, Supplemental Information). Brownstein (2019) thus misinterpreted two distinct and unrelated sentences in Cau et al. (2017), one about the peculiar features of the caudal vertebrae (not related to tail elongation/reduction), and another about the pelvic and femoral adaptations supporting hip-extension.

Brownstein (2019) wrote: “Therefore, the cross-sectional limb morphology of Halszkaraptor provides among the strongest evidence against a partially marine ecology in H. escuilliei”, and “[t]hese results were used to support a semiaquatic ecological mode in the taxon, with the forelimb acting as a propulsion device. However, the inferences made by Cau et al. [Ref. 32] from the morphometric analyses are flawed, as the forelimb of Halszkaraptor looks strikingly unlike the paddles formed by the forelimb bones of plesiosaurs” (Brownstein, 2019). Brownstein (2019) did not provide any quantitative morphometric analysis in support of his sentences. Contra Brownstein (2019), the cross-section geometry of Halszkaraptor’s ulna reflects an unusual flattening of the bone (a feature that was first noted in the other halszkaraptorine Mahakala, see Turner, Pol & Norell, 2011), and recalls the analogous condition differentiating wing-propelled aquatic birds from other avians (Simpson, 1946). When plotted relative to ulnar length, the mid-shaft mediolateral diameter of Halszkaraptor ulnar shaft clusters it among wing-propelled birds and not among other bird groups, and also results proportionally more expanded transversally than in other non-avian theropods (Fig. 4): this morphometric feature is consistent with the ecomorphological scenario of Cau et al. (2017). Brownstein (2019) also stated: “[…] this taxon [Halszkaraptor] was probably not biomechanically suited to live in water, as its skeleton, like other paravians, would have probably been too light to keep the animal submerged”, and “[t]he bones of Halszkaraptor are clearly internally hollow to a similar extent as other paravian dinosaurs. However, in tetrapods adapted for a semiaquatic or entirely aquatic lifestyle […], pachyostosis, the extreme thickening of cortical bone, occurs in the limbs. Given that pachyostosis is present in the limb bones of both avian and non-avian theropods that took to the water, the absence of such thickening in Halszkaraptor, which Cau et al. [Ref. 32] posit was well-adapted for a semiaquatic ecology, would be very surprising from a biomechanical standpoint” (Brownstein, 2019). In the above mentioned statements, Brownstein (2019) challenges Cau et al. (2017) arguing that vertebrates with hollow long bones and a highly pneumatized postcranial skeleton could not be adapted to some aquatic lifestyle, and implicitly claims that pachyostosis is a necessary requisite for a semiaquatic lifestyle. Both Brownstein’s (2019) assumptions are falsified by several modern birds, for example, the pelicans, characterized by an extensively-pneumatized skeleton (Richardson, 1939), and that are nonetheless well-adapted to piscivory, to exploit the aquatic environment and to a wing-propelled swimming style (Hinić-Frlog & Motani, 2010). It is noteworthy that the degree of internal bone cavitation and pneumatization in the skeleton of pelicans (Simons & O’Connor, 2012, fig. 3; Wedel, 2014; Wedel, 2018) is more extensive than in Halszkaraptor.

Figure 4 Plot of ulna mid-shaft width relative to ulnar length in theropods.

(A) Full sample. (B) Same sample but reduced to non-avian theropods and wing-propelled birds. Data in Supplemental Files.

Brownstein (2019) then questioned the analysis of morphospace occupation of Cau et al. (2017) which focused on the proportions of the medial fingers (I–II–III) in reptiles. He wrote: “Halszkaraptor lacks the “paddle” in plesiosaurs, Araripemys, and other aquatic vertebrates like ichthyosaurs, wherein the hand contains many closely appressed phalanges. In contrast, the forelimbs of marine reptiles, such as mosasaurs, plesiosaurs, and ichthyosaurs, consist of a massive number of flattened, heavily modified phalanges that form a distinctive paddle shape entirely distinct from the theropod manus”. Both aquatic chelonians and penguins show that a flipper- or paddle-like shape could evolve without hyperphalangy (Simpson, 1946; Walker, 1973; Clark & Bemis, 1979; Carpenter et al., 2010). The hands of wing-propelled birds have only three fingers, with a phalangeal formula even more reduced than in Halszkaraptor (Simpson, 1946). Thus, different skeletal morphologies may produce a functional paddle, which is not constrained to a five-fingered pattern and to hyperphalangy. To test if the hand of Halszkaraptor fits the overall proportions of a paddle, Cau et al. (2017) compared the proportions of the three medialmost fingers (fingers I–II–III) in reptiles. These fingers define the outline of the leading edge of the paddle, which is a key parameter in any flipper morphology (Combes & Daniel, 2001). The morphometric analysis showed that (1) there is not significant overlap between theropods and other reptiles in finger proportions, (2) Halszkaraptor does not cluster among the other theropods, and (3) the outline of the medial/leading edge of the hand in Halszkaraptor is more similar to those of aquatic reptiles than those of the other theropods. Brownstein (2019) failed to explain why Halszkaraptor shows so unusual finger proportions: the finger proportions in Halszkaraptor are not plesiomorphic for Maniraptora, and are not shared with herbivorous or omnivorous theropods, and thus do not fit Brownstein’s (2019) hypothesis. Contra Brownstein (2019), the forelimb of Halszkaraptor markedly deviates from those of other dromaeosaurids (e.g., Deinonychus, Ostrom, 1969; Microraptor, Hwang et al., 2002) in several features, including the overall stouter proportions of the bones, the marked flattening of the ulna, the significant reduction of the size of the first finger, the presence of a more robust third metacarpal, and the significant elongation of the phalanges of the third finger: it is noteworthy that all these features differentiate the forelimb of wing-propelled birds (e.g., penguins) from other (i.e., non-swimming) avians (Simpson, 1946).

Brownstein (2019) wrote: “Their resultant reconstruction of the glenoid facing laterally in H. escuilliei is therefore also unsubstantiated”. The rationale for the inference of a laterally-facing glenoid in H. escuilliei is explained by Cau et al. (2017), where it is stated: “Although the fragmentary preservation of the pectoral region prevents a detailed reconstruction of forelimb range of motion, on the basis of phylogenetic bracketing, we infer that the glenoid in Halszkaraptor faces laterally, as it does in forelimb-assisted swimming tetrapods”. A laterally-facing glenoid is a paravian symplesiomorphy inherited by dromaeosaurids (Turner, Makovicky & Norell, 2012) and thus, in absence of contrary evidence and based on phylogenetic bracketing, it is the most plausible condition for Halszkaraptor. Given that such a feature is also an adaptation necessary for any form of forelimb-assisted swimming (Carpenter et al., 2010), the plesiomorphic glenoid condition of paravians which is assumed for Halszkaraptor is also a potential exaptation for a forelimb-assisted swimming style.

Brownstein (2019) wrote: “Cau et al. [Ref. 32] noted that the “sickle” claw on pedal digit II is heavily reduced in Halszkaraptor compared to other dromaeosaurids”. The above-mentioned statement misreports Cau et al. (2017), who instead wrote: “The second toe is half the length of the third, with a stout phalanx II-2 and a large falciform ungual, similar to those in other paravians” (Italics added here). Contra Brownstein (2019), when pedal ungual II size of Halszkaraptor is plotted against femur length (a frequently-used proxi of body size in theropod research), there is no significant difference between H. escuilliei, the other dromaeosaurids, and other basal paravians (Fig. 5).

Figure 5 Pedal ungual II size among paravians.

Plot of pedal ungual II length relative to femur length dismisses Brownstein’s (2019) claim that Halszkaraptor’s ungual is reduced compared to other dromaeosaurids. Data in Supplemental Files.

Inaccurate or unsupported references to other taxa

Brownstein (2019) wrote: “Members of basal clades in the Dromaeosauridae, including microraptorans and unenlagiines, also possess a large number (20+) of teeth in their maxillae” (Italics added here). All known microraptorans have less than 20 teeth in their maxillae, not more (Turner, Makovicky & Norell, 2012, Fig. 23; Xing et al., 2013; Pei et al., 2014). Note that assuming (erroneously) a larger number of maxillary teeth in microraptorans has significant implications for the number of maxillary teeth inferred at the root of Dromaeosauridae (see below).

Brownstein (2019) wrote: “The complete connection of the postzygapophyses by bone surface [as in Halszkaraptor] is present in the basal-most ornithomimosaur Nqwebasaurus and the basal-most therizinosaur Falcarius, and is present to a lesser extent in basal alvarezsaurs like Aorun and Haplocheirus, the basal ornithomimosaur Pelecanimimus, and the basal tyrannosauroid Guanlong.”

Contra Brownstein (2019), all the above-mentioned taxa bear distinct postzygapophyses not-completely merged medially, and show posteriorly-concave interzygapophyseal laminae excavated dorsally by the ligament fossa (Choiniere et al., 2014a; Zanno, 2010; Choiniere, Forster & De Klerk, 2012), and thus lack the autapomorphic complex of H. escuilliei.

Brownstein (2019) wrote: “Although it is clear that the prominence of the supratrochanteric process in Halszkaraptor is greater than in these unenlagiines, the supratrochanteric process in many anchiornithids is similarly developed [Refs. 91 and 92]”. Brownstein’s (2019) Refs. 91 and 92, that is, Godefroit et al. (2013a) and Godefroit et al. (2013b), do not show the supratrochanteric process in anchiornithids. Furthermore, both Eosinopteryx (Godefroit et al., 2013a) and Aurornis (Godefroit et al., 2013b, Fig. 6A) lack a prominent supratrochanteric process like that claimed by Brownstein (2019) (A. Cau, 2015, personal observation). The supratrochanteric process of the anchiornithids is no more developed in shape and extent than the tuber-like process present in other paravians (e.g., compare Aurornis, Fig. 6A, or Anchiornis, Hu et al., 2009, fig. S4B, with Rahonavis, Turner, Makovicky & Norell, 2012, Fig. 55B) and is much less prominent than in Halszkaraptor, where it forms a peculiar large shelf-like lateral projection overhanging the ilium (Fig. 6B).

Figure 6 Development of the supratrochanteric process in the paravian theropods Aurornis xui YFGP-T5198 and Halszkaraptor escuilliei MPC D-102/109.

(A) Pelvic region of the anchiornithid Aurornis in lateral view. Note that the left ilium is exposed dorsally, showing the thickness of the dorsal margin of the bone. (B) Pelvic region of H. escuilliei, in dorsomedial view. Note the prominent supratrochanteric process which overhangs the lateral surface of the ilium. Scale bars = 30 mm. Abbreviations: li, left ilium; pdm, posterodorsal margin; ri, right ilium.

Methodological weakness and non-reproducibility of the phylogenetic analysis

Brownstein (2019) provided a data matrix in the Supplemental Information of his article. Unfortunately, the phylogenetic results described in Brownstein (2019) could not be obtained using the provided data matrix (i.e., the topology resulted using that data set is identical to that in extended data fig. 10 of Cau et al. (2017). Even more puzzling is the list of characters that Brownstein (2019) claimed to form the diagnosis of the clade formed by halszkaraptorines and unenlagiines, that he obtained in his analysis. He wrote: “This clade is united by five characters: 27 (0, maxillary fenestra situated at anterior border of antorbital fossa), 107 (1, Sacral vertebrae number is six), 193 (1, ascending process of astragalus short and slender), 580 (0, sagittal crest of parietal comprised of two parallel crests), and 828 (0, Meckelian groove centered)” (Brownstein, 2019). Note that character numeration does not follow entirely the original character list (Supplemental Information of Brusatte et al. (2014)): character statements #27, #107, #580 and #828 in Brownstein (2019) are instead statement #28, #108, #581 and #829 in Brusatte et al. (2014). Three of the above-listed character states could not be unambiguous synapomorphies of the “Halszkaraptorinae + Unenlagiinae” node, because they are actually absent among halszkaraptorines. The maxillary fenestra is only known in Halszkaraptor among halszkaraptorines (Cau et al., 2017). In this taxon, it is placed posterodorsally on the antorbital fossa and not “at anterior border of antorbital fossa”: thus, character 28 of Brusatte et al. (2014) cannot be scored as “0” in any halszkaraptorine. The ascending process of the astragalus is only known in Mahakala among halszkaraptorines (Turner, Pol & Norell, 2011). In this taxon, the ascending process is wide and covers the whole anterior surface of the tibia (state 0 of character 193 of Brusatte et al. (2014)), and is not slender and restricted over the lateral half of the tibia as in the state “193.1” of Brusatte et al. (2014). The parietal is only known in Halszkaraptor among halszkaraptorines (Cau et al., 2017). In this taxon, the bone entirely lacks a sagittal crest, and thus, following the description of the character in Brusatte et al. (2014), character 581 is inapplicable in Halszkaraptor.

Phylogenetic test

The phylogenetic analysis performed here reconstructed >99.999 shortest trees of 6,566 steps (CI = 0.2333, RI = 0.5558). The agreement subtree topology (Fig. 7A) is formed by 158 of the included 185 operational taxonomic units, indicating a relatively stable and well-supported framework among the majority of the taxa. The relationships among the main coelurosaurian clades are in agreement with the previous iterations of this data set (Cau et al., 2017; Cau, 2018). The analysis supports the sister group relationships between halszkaraptorines and unenlagiines, as found in Cau (2018), Gianechini et al. (2018), Hartman et al. (2019), and advocated by Brownstein (2019). This clade is supported by 11 unambiguous synapomorphies (Supplemental Files). Note that the result of this analysis confirms only one of the five synapomorphies suggested by Brownstein (2019) in support of this clade (the presence of six sacral vertebrae). Several features discussed by Brownstein (2019) and claimed in H. escuilliei are falsified by a careful analysis of the morphology of Halszkaraptor and other taxa, and do not support Brownstein’s (2019) scenario (e.g., Halszkaraptor actually lacks the “low tooth-replacement rate”, a “short tail”, or a “reduced” second toe ungual). Other purported features listed by Brownstein (2019) cannot be considered as shared morphological character statements because the condition in Halszkaraptor is not topographically homologous to those in non-paravian maniraptoriforms (e.g., the “platyrostral” premaxilla of Halszkaraptor cannot be homologous to those in ornithomimosaurus or therizinosauroids). Once these features are removed from the list of phylogenetically significant features forming the Halszkaraptorine body plan, the latter is described by 17 morphological character statements (Table 1). Character state transition optimization indicates that the majority of the features (11 over 17) are evolutionary novelties acquired along the “Halszkaraptorinae + Unenlagiinae” clade after its divergence from the other dromaeosaurids (Table 1; Fig. 7A). Seven of these halszkaraptorine novelties are convergently acquired by spinosaurids (Table 1). Contra Brownstein’s (2019) scenario, only two among the 17 features discussed (i.e., the absence of serration in the premaxillary dentition, and the presence of a robust metacarpal III) are maniraptoromorph and paravian symplesiomorphies, conserved in Halszkaraptor and lost along the “microraptorine + eudromaeosaurian” lineage.

Figure 7 Phylogeny of the tetanuran theropods focusing on maniraptoriforms.

(A) Agreement subtree of 50.000 shortest trees reconstructed by the phylogenetic analysis, used as framework for character state transition optimization. Numbers at branches indicate the morphological features listed in Table 1. (B) Anagenetic distance (in steps) from the paravian node based on the minimum branch length of the agreement subtree in (A).

Table 1 Phylogenetic status of 17 key features of Halszkaraptor.

Nodal optimization of the morphological features of Halszkaraptor body plan, based on the agreement subtree topology. Character numeration refers to the character list of the phylogenetic analysis, with described state indicated by number in brackets. Ambiguously optimized state changes based on accelerated transformation (marked by *). “Novelty” means that the character state in H. escuilliei is optimized as evolving among Halszkaraptorinae or at most among “Halszkaraptorinae + Unenlagiinae” under accelerated transformation optimization.

Character statement and homoplasy index (hi)	#Char.	Nodes	Status in Halszkaraptor	
(1) Premaxillae fusion; hi = 0.833	11(1)	“Halszkaraptorinae + Unenlagiinae”*. Pygostylia.
Oviraptoroidea. Spinosauridae	Novelty. Convergent with pygostylians, oviraptoroids and spinosaurids	
(2) Premaxillary narial margin placed posterior to mid-lenght of premaxillary oral margin; hi = 0.941	27(1)	Averostra. Lost in: “Microraptorinae + Eudromaeosauria”*, Troodontidae, among jeholornithids, among ornithomimosaurs, in Allosauroidea, in Tyrannosauroidea	Averostran plesiomorphy. Note that Halszkaraptor shows a novel state: the narial margin placed more posterior then the whole premaxillary body (convergent with avialans and spinosaurids)	
(3) Number of premaxillary teeth >4; hi = 0.8	14(1)	“Halszkaraptorinae + Unenlagiinae”*. Ornithomimosauria*. Spinosauridae	Novelty. Convergent with basal ornithomimosaurs and spinosaurids	
(4) Premaxillary teeth unserrated; hi = 0.875	15(1)	Maniraptoromorpha. Homoplastic in dromaeosaurids and troodontids. Spinosaurinae	Maniraptoriform symplesiomorphy. Convergent with spinosaurines	
(5) Lateral teeth unserrated; hi = 0.917	159(1)	Alvarezsauroids more derived than Haplocheirus*. Pannaraptora*. Lost at Dromaeosauridae root*. Re-gained in “Halszkaraptorinae + Unenlagiinae”.
Homoplastic in Troodontidae. Spinosaurinae	Novelty. Convergent with non-dromaeosaurid pennaraptorans and spinosaurines	
(6) >20 maxillary teeth; hi = 0.9	34(1)	Maniraptoriformes. Lost in Pennaraptora. Re-gained in “Halszkaraptorinae + Unenlagiinae”. Re-gained in “Sinovenatorinae + Troodontinae”. Baryonychinae*	Novelty. Convergent with non-pennaraptoran maniraptoriforms and baryonychines	
(7) Premaxillary teeth incisiviform; hi = 0.917	16(1)	Averostra*. Lost in Alvarezsauroidea*, Averaptora and Oviraptoroidea*. Homoplastic in Eudromaeosauria and Troodontidae	Averostran symplesiomorphy	
(8) Lateral teeth labiolingually compressed; hi = 0.875	599(0)	Theropoda. Lost in Spinosauridae	Theropod plesiomorphy	
(9) Cervical vertebrae elongate (centrum more than twice longer than deep); hi = 0.875	222(1)	Halszkaraptorinae*. Fukuivenator. “Caudipterydae + Oviraptoroidea”*.
Lost in heyuannines*. Falcarius. “Deinocheiridae + Ornithomimidae”. Spinosauridae	Novelty. Highly homoplastic among other maniraptoriforms. Convergent with spinosaurids	
(10) Horizontally-oriented caudal zygapophyses; hi = 0	1726(1)	Halszkaraptorinae	Novelty	
(11) Prominent caudal prezygocostal laminae; hi = 0.857	626(1)	Neotetanurae. Lost in derived ornithomimosaurs, some alvarezsauroids, oviraptorids, and in Eumaniraptora. Re-gained in “Halszkaraptorinae + Unenlagiinae”	Novelty. Homoplastic among other maniraptoriforms	
(12) Robust metacarpal III; hi = 0.95	322(0)	Eumaniraptora*. Lost in “Microraptoria + Eudromaeosauria”*. Homoplastic among microraptorines.
Lost in “Balaur + Pygostylia” Lost among Anchiornithinae. Derived therizinosaurids	Eumaniraptoran symplesiomorphy	
(13) Elongate manual phalanx p1-III; hi = 0.933	292(0)	Lost in Tetanurae*. Re-gained in Microraptorinae*, Halszkaraptorinae*, Scansorioperygidae and Pengornithidae*	Novelty. Convergent with some paravian lineages	
(14) Shelf-like iliac supratrochanteric process; hi = 0.5	1,773(1)	“Halszkaraptorinae + Unenlagiinae”*, lost in Unenlagia	Novelty	
(15) Elongate posterolateral crest on femur; hi = 0.75	693(1)	Ceratonykini*. Halszkaraptorinae. Late-diverging troodontids*	Novelty. Convergent with a few maniraptorans	
(16) Markedly convex extensor surface of metatarsal III; hi = 0.5	1,616(1)	Halszkaraptorinae. Balaur	Novelty	
(17) Unconstricted proximal end of metatarsal III; hi = 0.941	483(0)	Theropod plesiomorphy. Homoplastically lost among alvarezsauroids. Homoplastic in Oviraptorosauria. Lost in Ornithomimidae, Microraptorinae, Unenlagiinae and Troodontidae	Theropod plesiomorphy	

Discussion

Cau et al. (2017) interpreted the body plan of Halszkaraptor as being adapted to a peculiar ecology, able to exploit both terrestrial and aquatic resources. A similar interpretation has been suggested for spinosaurids (Hone & Holtz, 2017). Both terms “amphibious” and “semi-aquatic” have been used differently in literature, referring to a very broad disparity of ecologies, and may be misleading (see discussion in Hone & Holtz (2017)). Here, I delineate the possible ecology of Halszkaraptor comparing its peculiar body plan with those of living taxa. Among extant tetrapods, the sawbills (Anseriformes, Mergini, Mergus spp.) are probably the closest ecological analogous to Halszkaraptor, due to a remarkable series of similarities. Sawbills are long-necked birds with an elongate and moderately platyrostral snout which bears a serrated edge (analogous to a toothed oral margin), used to catch small fish and invertebrates (Nilsson, 1972; Kear & Hulme, 2005). When moving on land, sawbills assume a distinctly erect (hip-extended) body posture (Kear & Hulme, 2005); on water, they are characterized by a swimming model including forelimb-propelled locomotion (Hinić-Frlog & Motani, 2010). This peculiar combination of features, distinctive of sawbills, has also been inferred for Halszkaraptor (Cau et al., 2017), and supports for the latter a piscivorous and aquatic ecology similar to that of the mentioned avian clade.

The quality of the arguments provided in Brownstein (2019) in order to dismiss the main conclusions of Cau et al. (2017) is dramatically weakened by a long list of inaccurate reports, mostly due to the misinterpretation of the anatomical traits and bibliography. As shown above, most of this rebuttal paper has been necessarily devoted to identify and correct all these problematic statements, most of which are fundamental in Brownstein’s (2019) alternative scenario, and to remove them from the proper comparison of the two hypotheses. Brownstein (2019) misinterpreted several sentences in Cau et al. (2017) and thus provided a largely inaccurate and misleading depiction of the latter. Several statements that Brownstein (2019) referred to Cau et al. (2017) are actually absent in the latter. The absence of polydactyly and the lack of pachyostosis in Halszkaraptor are not valid arguments challenging the evolution of a semiaquatic ecology, because several tetrapod lineages (including some wing-propelled diving birds like pelicans) evolved such an ecology in absence of those anatomical features. Given that Cau et al. (2017) did not suggest a “partially marine ecology” for Halszkaraptor, and did not suggest a plesiosaur-like locomotory style or a plesiosaur-like forefin morphology in Halszkaraptor, it is unclear why Brownstein (2019) had focused to that peculiar fully-aquatic bauplan. Note that all aquatic and diving birds (both flying and flightless) lack the plesiosaur-like features in the forelimb listed by Brownstein (2019), so the absence of a plesiosaur-like paddle or a plesiosaur-like swimming style do not necessarily invalidate locomotion in water or a semiaquatic ecology in a maniraptoran theropod. Note that Cau et al. (2017) described the locomotory style of Halszkaraptor using the relatively neuter term “forelimb-assisted swimming” instead of any stronger term that may indicate a peculiar locomotory style more closely analogous to those of, for example, penguins or plesiosaurs. Thus, contra Brownstein (2019), focusing on the absence of fully-aquatic adaptations in Halszkaraptor does not affect the arguments discussed in Cau et al. (2017). Brownstein (2019) suggested that most of the features forming the unusual body plan of Halszkaraptor are maniraptoriform or maniraptoran plesiomorphies which were subsequently lost along the lineage leading to Eudromaeosauria. As shown above, a significant part of the features listed by Brownstein (2019) in support of that hypothesis are not valid, being based on inaccurate reports not supported by the literature cited therein. In most cases, those statements are based on misinterpretation of the anatomical terminology, or are grounded on problematic homology statements. In the most problematic cases, the mention of those features is merely false, being them absent in the holotype of H. escuilliei (e.g., the so-called “dentary chin” is not present in MPC-D 102/109). Once tested quantitatively, the remaining character statements mentioned by Brownstein (2019) are in large part inferred as synapomorphies of the halszkaraptorine lineage or, at most, as synapomorphies of the clade also including the unenlagiines (Gianechini, Makovicky & Apesteguía, 2011, 2017; Gianechini et al., 2018), and were acquired by that lineage after its divergence from the other dromaeosaurids. Contra Brownstein (2019), the most parsimonious scenario places the loss of serration in the lateral dentition, the increased number of lateral teeth, the elongation of the neck, and the development of the prominent supratrochanteric shelf, as novelties acquired along the “halszkaraptorine-unenlagiine” lineage: all these features were not inherited from maniraptoriform ancestors, and were not secondarily lost in eudromaeosaurs. The majority of the similarities with some maniraptoriforms are homoplastic convergences (a phenomenon widespread among theropod dinosaurs, see Holtz, 2001). At least seven of the halszkaraptorine novelties are convergently acquired by spinosaurids, and are integrated in a semiacquatic and piscivorous ecology (Charig & Milner, 1997; Ibrahim et al., 2014; Cau et al., 2017; Table 1). One of these features, reported here for the first time, is the “festooning pattern” in the upper dentition size variation, which recalls semi-aquatic crocodilians (Charig & Milner, 1997; Dal Sasso et al., 2005; Pol, Turner & Norell, 2009). In Halszkaraptor, the anteriormost two maxillary teeth (and corresponding alveoli) are smaller and much slender than the other anterior maxillary teeth and also smaller than the largest premaxillary teeth: this condition produces a distinct sinusoidal (“festooning”) oral margin due to the presence of two zones bearing elongate fang-like teeth, one in the premaxilla and one in the anterior half of the maxilla, separated by a zone bearing reduced teeth (Fig. 3). This condition is markedly different from the straight and uniform cutting surface present in the oral margin of the herbivorous theropods (Zanno & Makovicky, 2011; see Fig. 1A), and has been interpreted as an adaptation for foraging efficiently in aquatic environments and for grabbing evasive prey items (Vullo, Allain & Cavin, 2016). This snout morphology is frequently associated with the presence of numerous neurovascular pits opening on most of the premaxillary surface (Vullo, Allain & Cavin, 2016), which is also shared by H. escuilliei and spinosaurids. Assuming that the peculiar halszkaraptorine features are maniraptoriform plesiomorphies (as claimed in Brownstein (2019)) is not the most parsimonious explanation of the evidence, because it would require the secondary loss of all these claimed ancestral states in oviraptorosaurs, in the “avialan-troodontid” lineage (Averaptora) and in the “eudromaeosaur-microraptorine” lineage. The scenario supported here confirms the hypothesis that, during their evolution, different coelurosaurian lineages converged to a non-ziphodont, multitoothed, and long-necked body plan independently each other (Zanno & Makovicky, 2011; Choiniere et al., 2014b). It is noteworthy that the result of the current study is obtained setting ambiguous character optimization to favor reversals over convergences (ACCTRAN optimization), and thus endorsing a possible “deep” (maniraptoriform) origin of the halszkaraptorine features and their later reversal among eudromaeosaurs (as suggested by Brownstein (2019)): even with that optimization, the majority of the discussed features are recovered as synapomorphies of the halszkaraptorine lineage, and cannot be interpreted as maniraptoriform plesiomorphies (contra Brownstein, 2019). Brownstein (2019) consistently re-interpreted most of the features of Halszkaraptor listed by Cau et al. (2017) as plesiomorphic conditions of clades more inclusive than Halszkaraptorinae: careful comparison of the terms used in the two papers shows that the character descriptions used by Brownstein (2019) differ from those in Cau et al. (2017) in not distinguishing neomorphic and transformational character statements (Sereno, 2007). For example, Brownstein (2019) did focus on the presence of the supratrochanteric process of the ilium (a neomorphic character state shared by many paravians and therizinosauroids) to challenge Cau et al. (2017), whereas the latter did discuss the development of the shelf-like supratrochanteric process (a transformational character state present uniquely among the halszkaraptorine-unenlagiine lineage). As a consequence of such misinterpretation, what is a genuine apomorphy of the halszkaraptorines is erroneously claimed to be a maniraptoran plesiomorphy. Similarly, Brownstein (2019) did focus on the presence of the interpostzygapophyseal lamina in the cervical vertebrae (a neomorphic character state widespread among maniraptoriforms), whereas Cau et al. (2017) did discuss the development of the expanded and fan-shaped interpostzygapophyseal lamina in the cervical vertebrae (a transformational character state autapomorphic of H. escuilliei): the plesiomorphic status of the neomorphic state does not invalidate the autapomorphic status of the transformational one. It should be remarked that Brownstein’s (2019) scenario failed to provide an evolutionary explanation for the autapomorphic features that even the latter paper recognizes as being present in Halszkaraptor. The mere assertion of a “transitional” morphology in Halszkaraptor does not provide an explanation for its autapomorphies, because the latter, by definition, are not states intermediate between non-dromaeosaurids and later-diverging dromaeosaurids, but are instead novel features acquired uniquely along the terminal branch. All these features are unexplained under Brownstein’s (2019) scenario, because they are not maniraptoran plesiomorphies and are not correlated to an herbivorous/omnivorous ecology (Zanno & Makovicky, 2011). Given that these features are observed among piscivorous and aquatic amniotes, as discussed by Cau et al. (2017), and in absence of an alternative explanation for their presence in Halszkaraptor, the ecomorphological hypothesis discussed by the latter study keeps being valid even under the revised phylogenetic framework advocated by Brownstein (2019). Paradoxically, the sister-taxon relationship between Halszkaraptorinae and Unenlagiinae suggested by Brownstein (2019) (but see it discussed also in Cau (2018); Gianechini et al. (2018); Hartman et al. (2019)), weakens the so-claimed “transitional” status for the morphology present in Halszkaraptor, because it removes the latter taxon from a more direct basal divergence near the ancestral dromaeosaurid node, and places it nested among a disparate branch of non-eudromaeosaurian dromaeosaurids (Novas et al., 2009; Gianechini et al., 2018). Furthermore, the amount of morphological divergence of the halszkaraptorines from the ancestral paravian root is comparable to those of microraptorines and velociraptorines (Fig. 7B): asserting that Halszkaraptor is “likely representative of the morphological transition from the ancestral body plan of maniraptorans to the one [sic] that characterized dromaeosaurids” (Brownstein, 2019) is thus unjustified. In sum, even under the phylogenetic framework advocated by Brownstein (2019), there is no reason for assuming that the disparate morphologies represented by Halszkaraptor and the unenlagiines were “plesiomorphic” or “transitional” between the basal maniraptoran bauplan and other dromaeosaurids. The evolutionary scenario suggested by Brownstein (2019) is thus falsified by its own phylogenetic structure.

Conclusions

The hypothesis that the body plan of Halszkaraptor represents a “transitional” condition intermediate between non-paravian maniraptoriforms and eudromaeosaurians is based on a series of non-rigorous homology hypotheses, on the misinterpretation of several character statements describing the coelurosaurian diversity, and has been erected over a problematic list of literature misreports and misquotes. Halszkaraptor markedly diverged from the other maniraptorans, and careful investigation of the character state distribution among coelurosaurs confirms that the large majority of the peculiar features of H. escuilliei are not maniraptoran symplesiomorphies, and cannot define the ancestral dromaeosaurid body plan. A quantitative analysis of the morphological divergence among these taxa falsifies Brownstein’s (2019) scenario, dismissing a “transitional” status for the halszkaraptorines relative to other dromaeosaurids. Furthermore, that hypothesis is unable to interpret the peculiarities of the halszkaraptorines which are absent in the herbivorous/omnivorous maniraptoriforms, and fails to explain the similarities between Halszkaraptor, semiaquatic birds and piscivorous reptiles.

Supplemental Information

Supplemental Information 1 Phylogenetic data matrix in Nexus format and character list.

Data matrix in Nexus format describing the state distribution of 1807 character statements among 185 operational taxonomic units. Character statement list included at bottom of file.

Click here for additional data file.

Supplemental Information 2 Phylogenetic data matrix in tnt format.

Data matrix in tnt format describing the state distribution of 1807 character statements among 185 operational taxonomic units. Character list in Supplemental File 1.

Click here for additional data file.

Supplemental Information 3 Measurements data used for Figure 4.

Click here for additional data file.

Supplemental Information 4 Measurements data used for Figure 6.

Click here for additional data file.

I thank P. Godefroit for access to MPC D-102/109, and to V. Beyrand, P. Tafforeau and D. Voeten for rendering of scan data. I thank L. Zanno for the helpful information on the anatomical terminology, S. Lautenschlager for providing digital rendering images of MPC D-100/111, and to D. Iurino for providing digital rendering images of MSNM V4047. I thank Academic Editor F. Knoll, F. Agnolin, C. Brownstein and D. Hone for the revision of the first draft of this manuscript. The program TNT is being made available with the sponsorship of the Willi Hennig Society.

Institutional Abbreviations

MPC Institute of Paleontology and Geology, Mongolian Academy of Sciences, Ulanbatar, Mongolia

MSNM Natural History Museum, Milan, Italy

RBINS Royal Belgian Insitute of Natural Sciences, Brussels, Belgium

YFGP Yizhou Fossil and Geology Park, Yizhou, China

Additional Information and Declarations

Competing Interests

Author Contributions

Data Availability

The author declares that he has no competing interests.

Andrea Cau analyzed the data, prepared figures and/or tables, authored or reviewed drafts of the paper, and approved the final draft.

The following information was supplied regarding data availability:

The data matrix of the phylogenetic analysis and morphometric measurements are available in the Supplemental Files.

Halszkaraptor escuilliei holotype specimen is registered under the catalog number of the Institute of Paleontology and Geology (Mongolian Academy of Sciences, Ulanbatar, Mongolia): MPC D-102/109. It is provisionally housed in the collection of the Royal Belgian Institute of Natural Sciences (Brussels, Belgium).

CT-Scan data of Halszkaraptor escuilliei holotype is open access and available at the European Synchrotron Radiation Facility website page: http://paleo.esrf.eu/galleries/vertebrate_paleontology/dinosaurs/Halszkaraptor_escuilliei/org_slices/.

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
