# Peer review of "The body plan of Halszkaraptor escuilliei (Dinosauria, Theropoda) is not a transitional form along the evolution of dromaeosaurid hypercarnivory"

_PeerJ, doi:10.7717/peerj.8672_

## Round 0.1 · original submission · Major Revisions

Dear Dr Cau,

I have now received three reviews of your manuscript and invite you to revise and resubmit your manuscript, taking into account every point raised (and this includes providing a brief explanation of what you understand by 'semiaquatic').

Please, together with your unmarked revised manuscript, provide a marked-up copy as well as a document explaining how you have addressed each of the points raised by all three reviewers.

Best regards,
Fabien Knoll

·

Basic reporting

The manuscript fills the requirements for being considered for publication in PeerJ. It is concise and well-written, and only needs minor changes.

Experimental design

Methods are sufficient and accurate

Validity of the findings

The findings are totally supported by the evidence presented in the MS and include interesting novel data

Additional comments

I congratulate the author for suhc interesting contribution that sheds light on the poorly known and enigmatic theropod Halszkaraptor. It is a very valuable contribution

·

Basic reporting

See below. Although I commend the author for their speed in writing this response, I have many concerns about how this paper argues for a semiaquatic ecology in Halszkaraptor. The paper repeats many problems with the original paper, including improper comparisons of the anatomy of Halszkaraptor with aquatic and semiaquatic taxa that employ disparate locomotory modes and ecologies and are in fact very different in the relevant features of their anatomy. This paper also presents the arguments published in Cau et al. (2017) in a way that misleadingly supports the argument made here. I also have concerns about specific details of the arguments presented that the author considers factual but are actually not correct. These include statements made on the anatomy of the dentaries of dromaeosaurids, the forelimb of Halszkaraptor, and other features.

Experimental design

See below. Although I commend the author for performing new morphometric, phylogenetic, and optimization analyses, I have a number of major concerns about the over-interpretation of these results to fit the argument made in the paper. The author over-interprets the results of the optimization analysis, suggests the phylogenetic analysis conducted in the rebuttal was highly flawed even though it essentially matched the topology found in Cau et al. (2017) with the exception of finding a sister-taxon relationship between Unenlagiinae and Halszkaraptorinae (also found in the analysis here and in other analyses of the Cau et al. dataset). Many of the new morphometric analyses conducted are not relevant to the features the author discusses, such as the flattening of the ulna.

Validity of the findings

See below. I thank the author for clarifying a number of points in the original paper, but I also feel that this paper backtracks on the arguments made in Cau et al. (2017) in order to refute the rebuttal's arguments. I have concerns about the morphometric analyses, and believe the conclusions stem from important misinterpretations of the morphology of relevant taxa. Furthermore, the results of this study in some cases seem to advance arguments made in the rebuttal. Despite this, the author frames that study as highly flawed.

Additional comments

Peer Review of “The body plan of Halszkaraptor escuilliei (Dinosauria,
Theropoda) is not a transitional form along the evolution of
dromaeosaurid hypercarnivory”

Chase Brownstein

Dear Dr. Knoll,

I submit to you my review of Andrea Cau’s rebuttal of my recent rebuttal of a 2017 paper authored by him and colleagues in Nature. I commend the author for their quick, rigorous work, which is impressive. As laid out in my original paper, I have disagreements about what Halszkaraptor represents, but I very much agree that scientific discussion on this taxon is warranted and I appreciate and thank the author for putting together this paper responding to mine.

I am concerned with statements made in this rebuttal that repeat many of the errors made in the original paper with regards to the comparative anatomy work, the morphometric analyses conducted, and the phylogenetic analyses performed. I am also concerned that this paper backtracks on several arguments made in Cau et al. (2017) in order to characterize the rebuttal paper as inadequately responding to the original work's arguments. Finally, the many minor critiques presented here of the rebuttal paper neither serve to strengthen the argument about the ecology of Halszkaraptor (the goal of this paper) nor are in fact correct/supported by the anatomy of the relevant taxa.

Principally, this paper misconstrues the intent of the rebuttal. The transitional hypothesis was given as an alternative to the semiaquatic ecology hypothesis, which the rebuttal aimed to show was poorly supported. The way this paper is construed suggests the intent here is to suggest that if the alternative hypothesis (transitional form) does not hold, the semiaquatic hypothesis must. This is an incorrect assumption, as the null hypothesis in this case is that Halszkaraptor did not possess an aberrant ecology. Furthermore, this paper seems to confirm several points made in the rebuttal against interpreting the anatomy of Halszkaraptor to represent a semiaquatic specialization.

Given the issues I note, I think this paper in large part consists of critiques that do not serve to effectively support the original paper’s argument or show major flaws in the rebuttal. I do think there are plenty of important points made in this paper that would serve to strengthen the argument that Halszkaraptor is not transitional in some of the ways mentioned in the rebuttal. Note that this doesn't support a semiaquatic ecology in Halszkaraptor, which the rebuttal paper aimed to show was poorly supported.

Focusing on the following portions of the paper I think would provide an important response argument and contribution:

—Keeping the discussion of premaxillary expansion and foramina count increase.
—Conducting new morphometric analyses on the ulna of Halszkaraptor in line with the comments given above.
—Discussing the issue of the morphologies of the supratrochanteric process and cervical vertebrae
—Presenting the phylogenetic analysis and discussing why the rebuttal argument’s phylogeny is incorrect in important ways.

In the paragraphs below, I detail my concerns with particular sections of this paper.

Lines 71—93: Here, the author suggests that my characterization of the comparability of the rostral margin of the skull of Halszkaraptor with other maniraptoran theropods. Firstly, the paper presents my argument as one that is objectively inaccurate in the face of the literature. As the author discusses in lines 77—93, this is actually a question of anatomical homology, and so I think this section should be moved to the later general discussion in the paper. This section is predicted on the idea that anterior elongation and lateral expansion of the premaxillae is not homologous with the simple lateral expansion of the premaxillae. This is all well and good, but the point made in the original paper was that the expansion of the premaxillae observed in Halszkaraptor was unique to that taxon among Mesozoic paravians and most other coelurosaurs. The author seems to agree that the only major difference concerns the relative position of the internarial bar base along the anteroposterior margin of the premaxilla, because other coelurosaurs like therizinosauroids and ornithomimosaurs have laterally expanded premaxillae. That this feature is only more posteriorly placed than in other maniraptorans is a very tenuous feature on which to base comparisons with piscavorous archosaurs like crocodylians and spinosaurs. This is because spinosaurines like Spinosaurus and Irritator possess far more posteriorly retracted nares than Halszkaraptor (e.g., Ibrahim et al., 2014) and the rostra of crocodylians are far more flattened than theropods such that the nares/naris are/is strongly oriented dorsally and usually not visible in lateral view. Where the point of contention therefore lies is whether the premaxillary anatomy of Halszkaraptor is comparable to baryonychine spinosaurids like Baryonyx, which are known to have eaten fish (as well as a variety of terrestrial animals; Buffetaut et al., 2004). Using this feature to support the hypothesis that Halszkaraptor was a piscavore without considering these details is therefore premature, and the premaxillary margin of Halszkaraptor clearly shares more in common with the average maniraptoran than with clearly aquatic taxa like modern alligators, where the naris/nares are entirely oriented dorsally, or with spinosaurines, which show a variety of puzzling adaptations in their skulls and bones that may provide more evidence for piscivory or specialization for hunting aquatic prey. Although one might bring up the existence of putative ‘webbed’ baryonychine tracks (Theroplantigrada), the assignment of these tracks to baryonychines and the presence of authentic webbing on them has been characterized as dubious in the literature (Conti et al., 2005). Resultantly, there is little direct evidence to suggest baryonychines were heavily adapted for a semiaquatic lifestyle in the way Halszkaraptor was posited to be (e.g., as a forelimb-propelled or undulating swimming semiaquatic piscivore), and the two lineages differ in several key aspects of the features Cau et al. (2017) used to compare them.

95—105: This discussion concerns the author’s assertion that I misinterpreted the phrasing in Cau et al. While I agree there is a clarification to be made that the premaxillary morphology suggests that the naris is retracted, my point was to say that the naris might not extend so far posteriorly so that it would be allied with the condition in spinosaurids, to which Cau et al. (2017) originally compared Halszkaraptor. In fact, the skull reconstruction published in Cau et al. (2017) reconstructs the narial margin of the nares as being greater in anteroposterior length than the narial margin of the premaxillae. As I noted in my paper, this is not the condition seen in any dromaeosaurid for which the nasals and premaxillae are known. This statement is based on firsthand observation of the skulls/skull material casts of a variety of dromaeosaurids, including Bambiraptor, Velociraptor, Deinonychus, Dromaeosaurus, Tsaagan, etc.
With a quick look at the skulls of Velociraptor, Microraptor, or Deinonychus, this pattern becomes clear. The narial margin of the nasals is about the same length or far shorter than the margin in the premaxillae. If Cau et al. (2017) were to sufficiently compare this feature to the construction seen in baryonychine spinosaurids, they would have to argue the nares stretched far posteriorly, as is the case for Baryonyx and Suchomimus (Charig and Milner, 1997; Sereno et al., 1998). The nares of these spinosaurids are massive and stretch up to more than a fifth of the total length of their skulls. This is not the case in Halszkaraptor as the figures in this paper and in Cau et al. (2017) show. Therefore, the critique of the rebuttal paper’s argument presented here is misconstrued, because the naris would have to be elongate in addition to posteriorly located on the premaxilla to show a strongly retracted condition comparable to baryonychine spinosaurs.

106—120: As with many other statements in this rebuttal, this paragraph suggests that I misinterpreted the argument of Cau et al. (2017) that the bizarre rostral adaptations of Halszkaraptor are linked with a semiaquatic life or a life at least spent foraging for aquatic prey. See my other comments for why I feel this argument misconstrues what was written in the original study.

123—154: This section concerns the relative amount of neurovascular foramina along the premaxillae of Halszkaraptor relative to other taxa, and provides an important additional set of comparisons supporting the number of medially- and dorsally-placed foramina on the premaxillae of Halszkaraptor as a potential autapomorphy of this taxon. However, the author admits that other maniraptorans also do possess foramina on the dorsal surface of their premaxillae, something that was not said in the original paper. Given the scarcity of well-preserved premaxillae for basal members of many maniraptoran and coelurosaurian clades, the notion that the condition in Halszkaraptor of these foramina being more numerous is extremely unlike other maniraptorans seems a premature one to make. Although I agree that this feature seems potentially distinctive, its contribution to the ecological arguments given by Cau et al. should not be overstated. As the author notes and as is visible in theropods, superficial neurovasculature can change noticeably from taxon to taxon. Although this feature might be allied with that seen in crocodylians, there is not any strong evidence for that interpretation being better than the null hypotheses that this feature (1) is not highly distinctive given the poor basal paravian/maniraptoran record or (2) is unrelated to a piscivorous ecology.

156—190: The author here argues that I misrepresented the affinity of the tooth replacement condition in Halszkaraptor to those of other maniraptorans. From the acknowledgements, it is clear the author consulted with one of the authors of the paper I cited the most in my discussion of this feature. I very much appreciate that the author did this. Making sure to clarify homology is important but often overlooked work, so good on the author for doing so! However, I do not think the author’s discussion of this issue is exactly correct. As the author notes, there is a difference between the replacement pattern observed for the premaxillary teeth with the pattern observed for maxillary crowns in Halszkaraptor. Zanno and Makovicky (2011) discussed a generally low replacement rate as potentially related to herbivory in maniraptorans. Cau et al. (2017) state that the “protracted replacement in [the] anterior dentition” of Halszkaraptor was indicative of a semiaquatic ecology. This indicates that the replacement of the anterior dentition took a longer time than that of the maxillary dentition, which I allied with the delayed replacement seen in the teeth of maniraptorans in my paper. My argument was simply to say that the difference in maxillary/premaxillary replacement in Halszkaraptor might reflect that one of these changes in replacement was occurring.

192—200: The texts original 2017 paper clearly shows that comparisons with long-necked marine birds were made deliberately. The paper did not compare the neck of Halszkaraptor to giraffes, for example, because the point was to compare the cervicals of Halszkaraptor to semiaquatic and/or piscivorous species that also had elongate necks. The following quotations from the 2017 paper support this being the original argument:

“Compared to body size, the neck is elongate and forms 50% of the snout–sacrum length; this is the highest value found among Mesozoic paravians ….The first five neural arches are also
unique among theropods in their lack of inter-postzygapophyseal spaces: instead, each pair of postzygapophyses forms a single planar surface that faces ventrally and has a convex posterior margin. This morphology is also seen in some long-necked chelonians and a few birds (for example, Cygnus, Fig. 3h–j).” Cygnus is a long-necked aquatic bird.

Figure 3 caption in Cau et al. (2017) states: “Dorsal view of anterior cervical vertebrae of Cygnus (h), the fresh-water chelonian Araripemys (i) and Halszkaraptor (j). These vertebrae share a combination of features: (1) elongate neural arches with reduced ridge-like neural spines; (2) merged postzygapophyses that form a lobate process; (3) ribs fused to vertebra; and (4) horizontally oriented zygapophyseal facets.”

“Neck elongation is widespread among sauropsids that use an ambush mode of predation in water, and the cervical morphology of Halszkaraptor (unique among non-avian theropods) is exclusively shared with semiaquatic lineages such as araripemydid turtles and some long-necked anatids (Fig. 3g–i).”

The intention of the comparisons made was clearly to compare Halszkaraptor with long-necked semiaquatic birds and other long-necked semiaquatic lineages, including some turtles and plesiosaurs. It is therefore a mischaracterization of the original study in suggesting these discussions did not take place, and one that benefits the author’s stance that the rebuttal manuscript was picking at straws, so to speak.

211—217: The citation refers to what the sentence says about Halszkaraptor being the first aquatic non-avian maniraptoran, which is what the Cau et al. paper claimed. This paragraph is just a minor critique of where the reference was placed. Furthermore, the phylogenetic topology of Cau et al. (2017), when considered alongside their semiaquatic hypothesis for halszkaraptorines, might be taken to indicated a semiaquatic ecology was developed early in dromaeosaurid development. In any case, this issue does not impact the larger arguments made in the rebuttal regarding the semiaquatic ecology proposed by Cau et al. 2017.

219—229: The author suggests I mischaracterized the Djadokhta Formation paleoenvironment and simultaneously agrees that semiaquatic species are rare. The author suggests that the presence of the crocodyliform Shamosuchus in the same ecosystem indicates a fauna adapted for an aquatic lifestyle, drawing anatomical similarities between Shamosuchus and Halszkaraptor in regarding rostral morphology. See above for my comments on the non-homoplastic nature of the snouts of crocodylians and Halszkaraptor. Overall, the aridity of the paleoenvironment would have made aquatic ecosystems rare. Given that the original description suggested Halszkaraptor was pretty strongly adapted to an aquatic environment (e.g., had a modified snout and neck for catching aquatic prey and a modified forelimb and body for swimming), this line of argument still stands strong.

231—256: The author suggests I misreported the anatomy of the distal dromaeosaurid dentary, and that dromaeosaurids did not possess chins. As I have personally examined the type specimens of a number of dromaeosaurids in institutions like the American Museum of Natural History and Yale Peabody Museum, I am sure that my identification of this feature is correct. Other authors, including Evans et al. (2013) and Xing et al. (2013), identify this feature in dromaeosaurids like Microraptor, Velociraptor, and Acheroraptor. Simply put, this paragraph presents false information. The “straightened” morphology of the dentary of dromaeosaurids refers to the fact that these animals lack dentaries that thin anteriorly to form a triangular outline, which is seen in troodontids. In Figure 3 of the original study, Cau et al. (2017) point out that the ventral margin of the dentary of Halszkaraptor is distally confluent with the rest of itself. Since the rest of the dentary clearly narrows slightly, this produces a small but noticeable rounded dentary end. Therefore, the author’s conclusion that I misinterpreted the anatomy of dromaeosaurids is unsubstantiated.

258—263: The term ‘ziphodont’ has been used in the literature dealing with theropods to both refer to serrated and unserrated labiolingually compressed, recurved teeth with distinct carinae. Hendrickx et al. (2015) characterize ziphodonty in theropod dinosaurs to refer to “strongly labiolingually narrows crown[s] with a distal curvature, typically serrated carinae, and no constriction at the cervix,” and note that “if the large majority of ziphodont theropods show serrated teeth, some of them, such as Buitreraptor and Compsognathus, whose teeth do not always bear denticles, are still considered to have a ziphodont dentition.” Halszkaraptor clearly shows the unserrated condition, and so “unserrated ziphodont teeth” is a valid description. Critiquing my paper for using this term the way it was used is also irrelevant to the argument the author makes. The important result in the original paper was that the general morphology of the dentition of Halszkaraptor was communicated. This and other minor points of contention raised in this paper do not respond to the rebuttal paper’s arguments.

272—288: The skeletal reconstruction of Halszkaraptor given clearly shows that the original hypothesis for the caudal series of this taxon was that it was shorter relative to the body of H. escuilliei than the caudal series of some other dromaeosaurids (e.g., Velociraptor). The author seems to agree with what I said in my paper that the caudal series of Halszkaraptor is similar to those of basal members of other paravian lineages, and so saying that my paper was mistaken on this point is itself incorrect. Cau et al. (2017) state:

“The hyper-elongate neck of Halszkaraptor, countered by a less-elongated tail, suggests that its centre of mass was shifted anterior to the hip region.”

This statement and the skeletal reconstruction directly contradict the statement made in this paper that “ Cau et al. (2017) did not write that the unusual features in the caudal vertebrae of Halszkaraptor support a modified posture like that in birds: the latter was inferred on the basis of hypertrophied origin and insertion of the m. ileofibularis in, respectively, ilium and femur (Cau et al., 2017, supplementary information). Brownstein (2019) thus misinterpreted two distinct and unrelated sentences in Cau et al. (2017), one about the peculiar features of the caudal vertebrae (not related to tail elongation/reduction), and another about the pelvic and femoral adaptations supporting hip-extension.”

209—306: The author critiques my paper for not providing any morphometric analyses to back up my statement that the morphometric analyses presented in Cau et al. (2017) were flawed. The major issue here is that Cau et al. (2017) claimed that their morphometric analyses allied the forelimb of Halszkaraptor with those of plesiosaurs. As is shown in one of the figures in my paper, the forelimb of Halszkaraptor looks absolutely nothing remotely like those of plesiosaurs, so any morphometric analysis that found the two to be similar based on proportions of the particular long bones is inherently flawed. To illustrate this further, a human humerus and the humerus of some species of bat might have a similar length to width ratio, but just because this means they plot together in a morphometric analysis does not by any means say that the biomechanics of the bones are similar. This argument presented in Cau et al. (2017) for the functional anatomy of Halszkaraptor is supported by no morphological evidence.

Another issue is that the “unusual flattening” of the ulna in Halszkaraptor that the author claims is supported by the morphometric analyses conducted in the original paper and in this one is not in fact supported by any of these analyses. The two morphometric analyses conducted herein only plot mid-shaft width against total ulnar length, which could be taken to suggest something about the mid-shaft robusticity of the ulna but not anything about how this bone is flattened. A third variable factoring in both mid-shaft width measurements would need to be measured to get the morphometric analyses to support the arguments made.

The author's claim that my critique of this section of the original paper is invalid because I did not conduct morphometric analyses and instead relied on the gross dissimilarity of the forelimb of Halszkaraptor to that of plesiosaurs or wing-propelled birds is not supported. It is important to note the forelimb of wing-propelled birds is highly simplified relative to that of Halszkaraptor. The forelimb of a penguin, for example, is simply not highly comparable to Halszkaraptor, which has the classic three-fingered, clawed manus and elongate humerus, radius and ulna seen in non-avian coelurosaurs. Finally, the author fails to mention that the long axis of the ulna lies perpendicular to the vertical axis of the holotype specimen, along which it may have been deformed. On the whole, there is no strong evidence presented by this study or Cau et al. (2017) supporting a highly distinctive ulnar morphology in Halszkaraptor.

308—328: This section also attempts to discredit the rebuttal on the basis that it refutes statements that are irrelevant to inferring an aquatic ecology in Halszkaraptor. Although the author correctly points out that pachyostosis is not necessary for an aquatic ecology, the extreme adaptations for an aquatic life in Halszkaraptor, which are compared to those seen in crocodylians, aquatic turtles, and aquatic birds like Cygnus, mean that this aspect of the anatomy of Halszkaraptor ought to be considered. In aquatic birds like penguins and in crocodylians, a thickened layer of compact cortex along the periphery of bones is present (Houssaye et al., 2016). The comparison was therefore warranted. Furthermore, the comparison made between Halszkaraptor and pelicans is unjustified for several reasons. Pelicans and other flying birds possess a number of adaptations in the forelimb. The forelimb of pelicans is simplified and shows far more fusion than that of Halszkaraptor. These issues also confound direct comparison between Halszkaraptor and other marine birds. Halszkaraptor was suggested to possess an undulatory and/or forelimb propelled swimming mode, so the comparison made in this paper with pelicans mischaracterizes what was said in the original study. The author mischaracterizes an argument presented in the rebuttal by saying I said “vertebrates with hollow long bones and a highly pneumatized postcranial skeleton could not be adapted to some aquatic lifestyle, and [that I implicitly claim[s] that pachyostosis is a necessary requisite for a semiaquatic lifestyle.” The comparisons made were meant to go above and beyond what was originally said in Cau et al. (2017). So, the point made by the author here is irrelevant to the discussion.

330—350: The morphometric analysis and discussion presented by Cau et al. (2017) shows they argued the manus of plesiosaurs was similar to Halszkaraptor. Simply put, these skeletal elements look nothing like each other, and the assertion by the author that there is any morphological similarity between the two flies in the face of comparative anatomy. The suggestion that the anteriormost phalanges dictate the essential nature of the functional morphology of the manus is demonstrative that this section aims to defend a very poorly construed argument. Simply put, the manus of Halszkaraptor is not paddle-shaped, nor does it display hyperphalangy or extensive fusion and simplification as in two of the semi-aquatic and aquatic lineages with which it was explicitly compared in the morphometric analyses conducted by Cau here and in Cau et al. (2017). The manus of Halszkaraptor includes three digits tipped with sharp, curved unguals, the morphology seen in all other coelurosaurs. That the third finger is slightly aberrant is not indicative of anything, given that some of the few basal members of other derived coelurosaur clades we know of also possess bizarre manual digit proportions and digit proportion changes are well-documented among almost ever coelurosaur clade.

351—360: See the points made above.

362—374: This paper and Cau et al. (2017) are almost entirely focused on showing the anatomy of Halszkaraptor deviates markedly from other dromaeosaurids and paravians. Assuming that this argument is correct, there should be no reason to infer phylogenetic bracketing will tell us anything useful about the forelimb. This is picking and choosing which features ought to be reconstructed as distinctive or not distinctive. This is an especially important point to make given that the author considers the forelimb of Halszkaraptor to be highly aberrant rather than representative of several evolutionary transitions, although this is also not strongly supported by the anatomical evidence at hand.

376—384: I appreciate the use of morphometric analyses to support this point. However, a comparison of the foot of Halszkaraptor with the foot of a derived dromaeosaurid like Deinonychus shows the pedal ungual II of the latter was far larger and more developed relative to the other unguals. The morphometric analysis performed just shows that there is a linear relationship between log femoral length (a proxy for body size) and log ungual length. This is simply expected, as bigger animals will have bigger bones. What the author fails to do is compare the proportions of II-3 with the rest of the foot, which is the area of interest. The hypertrophied nature of dromaeosaurid pedal ungual II-3 primarily has to do with its size and extremely curved morphology relative to the other unguals of the foot. The plot seems to actually support the larger size of the pedal ungual II-3’s of derived dromaeosaurids compared to other paravians, especially if one remembers that a non-log-transformed version of this plot would show far more distance between the bin consisting of Halszkaraptor and other basal dromaeosaurids and paravians and that consisting of derived dromaeosaurids. The pedal ungual II of Halszkaraptor is also clearly not as heavily recurved as the same bone in more derived dromaeosaurids, including eudromaeosaurs.

432—459: The author suggests the data matrix I provided did not lead to the same phylogenetic topology I found when tested, yet I could not find in the supplementary information the tree that shows this. In any case, the phylogenetic topology I found in my rebuttal does not really differ at all from the topology found by Cau et al. (2015) or Cau et al. (2017), so I am unsure what dramatic problem the author has with the results I presented in my rebuttal of Cau et al. (2017). Furthermore, the author recovers the same sister-taxon relationship between Unenlagiinae and Halszkaraptorinae in the dataset they use. So, there is pretty much no distinction between the results I presented and the results presented here as far as the phylogenetic resolution and placement of Halszkaraptorinae is concerned, and this section, which claims to disprove my phylogenetic results, in fact supports them by finding the key relationship between Halszkaraptorinae and Unenlagiinae.

474—483: See above for why these arguments are not justified. None of the codings of Cau et al. (2015) and Cau et al. (2017) were changed in the analysis of Brownstein (2019), so the arguments made here do not in fact address the results of the latter study. If anything, the fact that very slightly different supports are found for particular clades in this matrix suggests the coding in Cau et al. (2017) may be sensitive such that slight differences in how analyses are run on the dataset produce significant differences in topology (although what major differences found are not clarified by the author here).

484—491: The author presents a list of 17 features that they state I mentioned in my original rebuttal or that they suggest are relevant to the development of the halszkaraptorine body plan. However, a careful review of these features suggests that the optimization results may actually be artifacts of the poor fossil record of basal members of many maniraptoran clades instead of synapomorphic to Halszkaraptorinae. The features discussed in my review include 2, 3, 4, 5, 6, 9, 14, and 17. Of these, only 3, 5, 6, 9, and 14 are found to be novel in halszkaraptorines, and 5, 6, and 9 are found to be homoplastic with respect to other maniraptorans and coelurosaurs. 14 should be considered a dubious character at best, because it describes the development of a process, and not the presence of the process itself. Although the development of the process might be important to the soft tissue anatomy in this region, the analysis fails to provide a justification for the level of aberrancy in this feature. Character 3 might be more strongly supported, although it is important to note that basal members of Therizinosauroidea and Oviraptorosauria are poorly characterized as of now, and little is known of the skull material of early members of these groups. The most basal therizinosaurs for which the premaxillae are known already possess edentulous premaxillae (Pu et al., 2013), and definite basal oviraptorosaurs already show highly specialized premaxillary dentition (Balanoff et al., 2009). Features 7, 8, and 12 are considered plesiomorphic with respect to maniraptorans and higher clades. In sum, the character state optimization results provide dubious support at best for the author’s assertion that many of the features of contention in Cau et al. (2017) and Brownstein (2019) are novel in Halszkaraptorinae. In this light, the paragraph starting on line 484 grossly overstates the results in the favor of the argument proposed by the author in this and other papers.

494—501: As noted in my comments above, this paper frames the rebuttal as incorrect based on
misrepresentation of the contents of the arguments presented in the original paper and my rebuttal. Many of the errors the author notes for my paper are actually either disagreements the author has with the homologous nature of features on Halszkaraptor, incorrect assertions, or discuss minor issues that do not affect the main arguments of the rebuttal study.

502—507: The statement made here is not reflective of what was said in Cau et al. (2017), who compared the limb anatomy to those of plesiosaurs using morphometric analyses, and suggested Halszkaraptor possessed a number of traits that they interpreted to represent adaptions for a piscivorous and aquatic lifestyle. Cau et al. (2017) provided two contradictory aquatic locomotory mode hypotheses for Halszkaraptor: forelimb-assisted swimming and undulation. Cau et al. (2017) state:

“The horizontally oriented zygapophyses in the neck and tail vertebrae of halszkaraptorines would have permitted the axial undulatory swimming mode that is typical of taxa with axially elongated body shapes”

Here and elsewhere in the original paper, Cau et al. (2017) repeatedly attempted to ally Halszkaraptor with fully aquatic or amphibious taxa in their study. Therefore, the author’s assertions in the discussion paragraph misrepresent the original paper in an attempt to discredit mine. Statements like “Cau et al. (2017) did not suggest such an extreme form of aquatic adaptation in H. escuilliei” and “Given that Cau et al. (2017) did not suggest a “partially marine ecology” [sic] for Halszkaraptor, and did not suggest a plesiosaur-like locomotory style or a plesiosaur-like forefin morphology in Halszkaraptor, it is unclear why Brownstein (2019) had focused to that peculiar fully-aquatic bauplan” are false, as Cau et al. (2017) state:

“The unusual morphology of Halszkaraptor suggests a semiaquatic ecology”

“The unusual forelimb morphology is not inconsistent with a semiaquatic ecology”

“Morphometric comparison of the forelimb of Halszkaraptor with those of terrestrial, aquatic and flying sauropsids supports the idea that this theropod possessed swimming adaptations”

“instead, the proportions of Halszkaraptor cluster with those of long-necked aquatic reptiles”

“Halszkaraptor is interpreted as an amphibious theropod: an obligatory biped on land and a swimmer that used its forelimbs to manoeuvre in water and that relied on its long neck for foraging”

The rebuttal never stated that Cau et al. presented Halszkaraptor as a fully aquatic animal, and referred to their hypothesis as suggesting H. escuilliei was partially marine or semiaquatic in ecology. This mischaracterization of the rebuttal, which does not say Halszkaraptor was fully marine or aquatic, and the original paper, which compares Halszkaraptor with fully and semiaquatic taxa, serves only to strengthen the assertion that the arguments presented in the rebuttal are flawed by attacking their relevance to the original argument. As the quotes above show, the rebuttal paper’s aims were indeed relevant, and the author’s words mischaracterize both what was said in the original paper and in the rebuttal to benefit their argument.

513—516: This sentence encapsulates a major issue with the original paper. In providing support for the hypothesis that Halszkaraptor was semiaquatic, Cau et al. allied the forelimb, cervical vertebrae, skull, and other bones of Halszkaraptor to semiaquatic and fully aquatic animals. My rebuttal attempted to show this comparisons did not ally Halszkaraptor with semiaquatic or aquatic taxa. Suggesting that just because some of these features are absent does not mean Halszkaraptor was not semiaquatic is equivalent to admitting the paucity of evidence for the ecological mode. The null hypothesis is that Halszkaraptor had the same ecology as any other theropod, whereas the author’s discussion assumes the null hypothesis is that Halszkaraptor is aquatic. This is an incorrect assumption, but is largely what this paper and the original one are predicated on.

524—541: See my previous comments on the strength of the discussion using the new optimization results.

541–543: Six of the seventeen features assessed are apomorphic to Paraves or larger clades, and a number of other features that the author suggests are homoplastic may reflect biases in the number of basal members of paravian lineages known. This sentence overstates the results.

Discussion:
This section provides an overview of what the author suggests was accomplished in this paper. Many of the points outlined above are relevant for understanding why this section incorrectly represents the issues of contention and the arguments made in the original paper and rebuttal.
As noted, many of the features the author says are strongly supported as peculiar synapomorphies of Halszkaraptorinae are in fact dubiously distinctive and/or poorly supported as independently developed in that clade. For example, whether the development of the supratrochanteric process is particularly important for the development of a semiaquatic body plan in Halszkaraptor is not supported, nor is the extreme aberrancy of this feature relative to other paravians. In large part, the discussion section backtracks from the original study in an attempt to mount an assault on the rebuttal.

The author presents a new feature, the size of the anteriormost premaxillary teeth, as further evidence of an aquatic ecology, comparing this feature to that seen in spinosaurids and semiaquatic crocodylians. The author misses that the "festooning pattern" where the anterior premaxillary teeth are much smaller is present in a huge number of other coelurosaurs, including tyrannosauroids (e.g., Sereno et al., 2009, Fig. 1), basal troodontids (Xu and Norell, 2004), basal alvarezsaurs (Choiniere et al., 2014, Fig. 12) and others.
The author is correct that the small size of the anterior premaxillary teeth of Halszkaraptor is unlike the condition in derived dromaeosaurids like Saurornitholestes or Velociraptor, but the widespread nature of this feature among clearly terrestrial coelurosaurs shows this feature is not at all indicative of a semiaquatic ecology. The author's attention to this anatomical feature is demonstrative of a major flaw in the arguments presented here. Principally, the widespread nature in Coelurosauria of many of the features in Halszkaraptor deemed indicative of a semiaquatic ecology in that taxon suggest these features are not drastically different from what is seen in terrestrial coelurosaurs. Many of these features may even be homologous and this study and the rebuttal show, and so there is no reason to say comparisons with semiaquatic taxa are indicative Halszkaraptor was semiaquatic when the very features discussed in these comparisons are actually present (in combinations in some cases) in many of Halszkaraptor's relatives along the coelurosaurian tree.

On the whole, this paper attempts to rebut my rebuttal of Cau et al. (2017) by claiming numerous minor issues supposedly exist within my paper that render its argument meaningless. As I’ve stated above, these critiques break down into two main types. The first includes minor statements that in most cases are predicated on the author’s own terminological definitions and assumptions about theropod anatomy, of which almost all are incorrect or poorly substantiated. The other include poorly construed arguments based on poor or simply absent morphological evidence. Although I think that some parts of this paper are worthy of publication, the majority of this work succumbs to these issues. However, I think many good points are also made in this paper, and I suggest the author focus on the one’s I’ve highlighted.

Regards,

Chase Brownstein
Research Associate,
Stamford Museum and Nature Center

References.
1. Cau, A. et al. Synchrotron scanning reveals amphibious ecomorphology in a new clade of bird-like dinosaurs. Nature 552, 395–399 (2017).
2. Ibrahim, N. et al. Semiaquatic adaptations in a giant predatory dinosaur. Science 345(6204), 1613–1616 (2014).
3. Conti, MA, et al. Jurassic dinosaur footprints from southern Italy: footprints as indicators of constraints in paleogeographic interpretation. Palaios 20.6 , 534-550 (2005).
4. Charig, A. J. & Milner, A. C. Baryonyx walkeri, a fish-eating dinosaur from the Wealden of Surrey. Nat. Hist. Mus. Lond. Bull. 53, 11–70 (1997).
5. Sereno, P.C., et al. A long-snouted predatory dinosaur from Africa and the evolution of spinosaurids. Science 282 (5392), 1298–1302 (1998).
6. Zanno, L. E. & Makovicky, P. J. Herbivorous ecomorphology and specialization patterns in theropod dinosaur evolution. Proc. Nat. Acad. Sci. USA 108, 232–237 (2011).
7. Evans, D.C., et al. A new dromaeosaurid (Dinosauria: Theropoda) with Asian affinities from the latest Cretaceous of North America. Naturwissenschaften 100(11), 1041–9 (2013).
8. Xing, L. et al. Piscivory in the feathered dinosaur Microraptor. Evolution 67, 2441–2445 (2013).
9. Hendrickx, C., et al. A proposed terminology of theropod teeth (Dinosauria, Saurischia). Journal of Vertebrate Paleontology, 35(5), e982797 (2015).
10. Brownstein, C.D. (2019). Halszkaraptor escuilliei and the evolution of the paravian bauplan.
Scientific Reports 9:16455.
11. Houssaye, A., et al. Adaptive patterns in aquatic amniote bone microanatomy—more complex than previously thought. Integrative and Comparative Biology, 56(6), 1349-1369 (2016).
12. Cau, A., Brougham, T. & Naish, D. (2015). The phylogenetic affinities of the bizarre Late Cretaceous Romanian theropod Balaur bondoc (Dinosauria, Maniraptora): dromaeosaurid or flightless bird? PeerJ 3, e1032.
13. Pu, H. et al. An Unusual Basal Therizinosaur Dinosaur with an Ornithischian Dental Arrangement from Northeastern China. PLoS ONE 8(5), e63423 (2014).
14. Balanoff, A. M., Xu, X., Kobayashi, Y., Matsufune, Y. & Norell, M. Cranial Osteology of the Theropod Dinosaur Incisivosaurus gauthieri (Theropoda: Oviraptorosauria). Am. Mus. Nov.3651, 1–35.
15. Sereno, P.C. et al. Tyrannosaurid skeletal design first evolved at small body size. Science 326 (5951): 418–422 (2009).
16. Choiniere, J. N., et al. Cranial osteology of Haplocheirus sollers Choiniere et al. 2010 (Theropoda, Alvarezsauroidea). Am. Mus. Nov. 3816, 1–44 (2014).

·

Basic reporting

No comment.

Experimental design

No comment.

Validity of the findings

The one area that I think can be improved is to provide some form of definition of the degree of aquatic lifestyle of Halszkaraptor. This causes issues both in this paper and in the commentary from the response paper meaning that there is a lack of clarity in some of the arguments when it's not obvious what the author means. A bit more on the details of the apparent convergences with spoinosaurs and adaptations to this lifestyle would also help. I appreciate these are not core to the central ideas of this paper but they do crop up repeatedly here and spelling out some of these arguments would be very useful here and going forwards.

Additional comments

Please see the marked up document for various suggested additions and corrections.

---

## Round 0.2 · accepted · Accept

Dear Dr Cau,
I accept your paper for publication. Given its nature, I strongly recommend that you make the full peer review history public so that the readers could have access to Mr Brownstein's point of view (although, of course, the decision to do so is yours alone).
Best regards,
Fabien Knoll

·

Basic reporting

See below.

Experimental design

See below.

Validity of the findings

See below.

Additional comments

Review of “The body plan of Halszkaraptor escuilliei (Dinosauria, Theropoda) is not a transitional form along the evolution of dromaeosaurid hypercarnivory” by Andrea Cau

Dear Dr. Knoll,

Thank you for allowing me to review this paper. I want to thank the author for rigorously responding to my previous comments and for clarifying/correcting when discussing particular features. However, I am still unconvinced of the findings, primarily because the author’s analyses either still do not in fact find fault with several of the points made in the rebuttal paper or themselves are, in my understanding, improperly performed for testing the anatomical hypotheses relevant to this debate. I apologize for being a bit difficult here, but I am still unconvinced by the findings presented in this and the original paper, and I feel as though many of the supposed similarities between Halszkaraptor and semiaquatic taxa are still poorly substantiated or not fleshed out. I do feel there might be a way forward for this paper, but not as it is currently framed. In short, there are serious issues with the inferences made here about comparative anatomy. Please see my detailed notes below for comments on some of the more important issues.

Firstly, the author continues to argue that the expansion of the rostrum in Halszkaraptor is sufficiently distinctive from other maniraptorans and coelurosaurs to suggest the feature in the former is not homologous with rostral expansion in members of those larger clades. The author differentiates the condition in Halszkaraptor into two features, the “remarkable anteroposterior elongation and dorsoventral flattening of the prenarial region of the premaxilla, which also results in the posterior placement of the narial region relative to the snout anterior tip.” Figure 2 clearly demonstrates, however, that the lateral expansion of the premaxillae found in therizinosaurs, ornithomimosaurs, and some other coelurosaurians is also present in Halszkaraptor. The author states in the rebuttal that the key feature here is the posterior retraction of the nares in Halszkaraptor, and so the issue is that the combination of these features produces the distinctive condition. Right now, both the author and I are in apparent agreement that at least half of this combination can be found widely among maniraptorans and coelurosaurs along the lineage to Paraves and within it, Dromaeosauridae. As I’ve noted in my previous review, I would be very, very cautious in inferring the distinctiveness of the posterior retraction of the premaxillary narial margin in Halszkaraptor. Looking at the holotype of Mei long, a very basal troodontid that we expect might look somewhat like basal dromaeosaurids based on the phylogenetic evidence, the narial margin of the premaxillae is posteriorly retracted (Fig. 2 in Xu and Norell, 2004). This may have to do with distortion of an apparently three-dimensionally preserved, complete specimen (reminiscent of Halszkaraptor, although the holotype of Mei shows arguably better preservation) (see Gao et al., 2012 for another specimen). So, not only is half of the combination noted here clearly not distinctive, but another key part of it might have to do with distortion or might even be present in a similar fashion in close relatives to basal dromaeosaurids. Repeat for the “dorsoventral flattening,” for which the author has not considered distortion as a factor. In any case, the apparent aberrancy noted for this part of the anatomy of Halszkaraptor remains unsubstantiated. I believe the author should talk about this feature, but only with the context of larger maniraptoriform anatomy.

Continuing with the discussion of narial retraction, the author states that there is “no a priori reason” for Halszkaraptor to show a narial morphology similar to eudromaeosaurs. The author here is clearly rebuttng my examples of eudromaeosaurs not showing the condition reconstructd for Halszkaraptor. The issue is that microraptorans also do not have a longer nasal margin of the naris than premaxillary margin (see Fig. 4 in Gong et al., 2012; Fig. 3 in Pei et al., 2014). Since we do not have unenlagiine skulls where this region is well-preserved yet, there is absolutely no phylogenetic support for the reconstruction made in Cau et al. (2017). The author notes that narial elongation is not relevant to the issue at hand, and rightly states that the posterior narial margin is not preserved for baronychine spinosaurs. Firstly, I would argue (as I have) that narial elongation is relevant, as this condition is clearly present in baronychine spinosaurs and might have to do with how soft tissue ought to be reconstructed for these animals. Without very close to exact morphological similarity in this feature between H. escuilliei and spinosaurids, I have a very tough time buying the argument made here. Furthermore, without any substantial support for the nasal condition in Halszkaraptor being substantially different from other maniraptorans and coelurosaurs (see here, above, and in the following discussion), there is not any scientific reason to compare the skull with spinosaurids. This is essentially “jumping the shark” when it comes to comparative anatomy/functional morphology, and here it is done without a shred of support from any biomechanical analysis. The nostrils of Halszkaraptor might have been slightly more posteriorly placed in life than other dromaeosaurids, but whether this condition reflects a substantial leap in ecomorphology (as the author suggests and has suggested based on their speculation of the locomotory abilities of Halszkaraptor) remains unsubstantiated.

The author suggests I contradicted myself by comparing the premaxillae of Halszkaraptor to the premaxillae of basal maniraptorans we do have and then suggesting the sample size of basal maniraptoran premaxillae was too small for such comparisons to be made well. As I stated in my previous review, the burden of evidence was on Cau et al. (2017) to provide conclusive comparisons demonstrating the strong aberrancy of features like the distribution of foramina on the premaxillae. I thank the author for clarifying that the combination of the distribution and number of foramina is what distinguishes Halszkaraptor. As the author notes, this is not exactly what was said in the original study. However, the point remains that the claim about the foramina count increasing on the premaxillae of Halszkaraptor requires far more material be found before this can be leaned on as a feature informative for discussions about that taxon’s ecology. In any case, claiming in this paper that Brownstein (2019) is blatantly incorrect for noting this feature is misleading.

The point about tooth replacement rates is well-taken. Thank you for clarifying this important distinction.

The author has claimed in their rebuttal paper that the reason that they only focused on comparing Halszkaraptor to long-necked marine reptiles and birds was because of cervical morphology, and not the elongate necks themselves. However, this point is not clarified at all in Cau et al. (2017). Cau et al. (2017) do state that the cervicals of Halszkaraptor are allied to the cervicals of long-necked marine taxa in several ways, but this comes under the implicit assumption made in that paper that long-necked taxa ought to be compared with each other. The author validates this point by saying they also examined giraffes. Furthermore, when Cau et al. (2017) presented morphometric analyses allying Halszkaraptor with plesiosaurs and other long-necked marine reptiles, the intention made was to compare Halszkaraptor to long-necked aquatic and semiaquatic taxa. This section in this paper therefore misrepresents what was actually said in the original paper, and improperly suggests that the rebuttal paper was at fault for discussing a feature that was incompletely discussed/explained in the original manuscript. The question of how the argument was advanced in Cau et al. (2017) is the one considered here, not about whether Halszkaraptor should or should not be compared with long-necked species. Furthermore, the authors of Cau et al. (2017) also failed to compare the elongated neck of Halszkaraptor with non-paravian coelurosaurs, as noted in the rebuttal and indirectly in this paper.

On the point about the absence of extensive lacustrine, fluvial, etc. environments fitting for an aberrant, semiaquatic, forelimb-propelled or neck-undulating swimming dinosaur, the author agrees that only one other vertebrate in the Djadokhta fauna shows any sign of being semiaquatic. Furthermore, the author fails to mention that Shamosuchus djadochtaensis
possesses an exceptionally short and narrow relative to other members of that genus, which is exactly the opposite of what is seen in baryonychine spinosaurs and Halszkaraptor . Nonetheless, the author goes on to describe several morphological similarities between Halszkaraptor and Shamosuchus as evidence of the semiaquatic ecology hypothesis for the former. This act is demonstrative of many superficial comparisons made in Cau et al. (2017) and this paper that do not accurately and effectively reflect the sum total morphological evidence. The author then goes on to cite a paper from the 1970s that reports lacustrine settings for part of the Djakhota. Please refer to my previous review and the citations in the rebuttal paper for a more recent, up-to-date discussion of Djakhota paleoenvironments. Given that the putative semiaquatic dinosaur Spinosaurus, which is the only other dinosaur besides Halszkaraptor for which supposed substantial adaptations for swimming have been described from the body and hindlimb, lived in a giant deltaic ecosystem, the question of the paleoenvironment of Halszkaraptor is still important to consider, and is not addressed in this paper with only a handful of at best dubious pieces of evidence.

On the issue of dentary bulge morphology, I can confirm that the feature reported in Evans et al. (2013) is the same I noted as a “chin.” The confusion here stems from the fact that, as Evans et al. (2013) report, this “chin” is poorly developed in Acheroraptor. It is heavily developed to form a bulge that also stems from this divergence in Velociraptor and Bambiraptor, as the figure in the rebuttal paper shows. Therefore, the author here misconstrued what was said in the original review. Since the morphology described for Halszkaraptor in the original paper is consistent with this feature in other dromaeosaurids, the note made in this paper is a clarification, and not a correction.

The issue of the rebuttal paper using the term “unserrated ziphodont” to describe the teeth of Halszkaraptor is not only a non-issue, but also misreports the etymology of the term, its history in the literature, and is employment in the rebuttal paper. This is done not out of a need to correct an egregious error in the original paper, given that unserrated theropod teeth have been described as ziphodont before (e.g., Hendrickx et al., 2015). Langston (1975) does indeed use the term, but he does not actually define it as having to do with serrated teeth! Ziphodonty in crocodylians has to do with whether their teeth are dinosaur-like (e.g., not conical). Furthermore, Langston (1975) provisionally credits Marsh (1871) with introducing the term. Brochu (2013) says that Langston (1975) said that serrated teeth are key to ziphodonty, but all Langston said is that Marsh was “impressed” with the recurved, mediolaterally compressed, and serrated teeth of some crocodyliforms. In fact, Langston and Marsh were using the term correctly, because, contra the author, ziphodont derives from a type of Iron Age sword with smooth edges described in Ancient Greek texts. Langston (1975) defines ziphodont on the first page of his paper as “the vernacular derived from the taxon ziphodon … proposed by O.C. Marsh for a species of dinosaur-toothed crocodilians from Wyoming. The term is descriptive of the principle distinguishing character state of this group of crocodilians and is less clumsy and more precise than the term “dinosaur-toothed” previously employed.”

So, the author here incorrectly characterizes the literature and repeats the incorrect use of this term in order to show a minor issue supposedly exists in the terminology of the rebuttal paper. Not only is this argument incorrect as I have shown it to be, but it also clearly is not scientific in origin. The term could be understood in the rebuttal paper even if serrations were integral to the term ziphodont as used in modern paleo literature. I can only assume that this section was included in order to demonstrate the inferiority of the rebuttal paper textually, which is not an attack on the science. The attempt to frame the rebuttal paper here as inaccurate because of small terminological issues unfairly benefits the author’s position on the issues at hand.

On the subject of the extension of distal end of the caudal series in H. escuilliei, it is clear from the published figures that, even though it is incomplete, the elongation of the tail in this taxon did not approach that seen in other dromaeosaurids and even other paravians. In the rebuttal letter, the author also notes the distinctiveness of some of the features of the caudals of halszkaraptorines relative to other dromaeosaurids. In any case, the fact that this feature is clarified as not being strongly involved in the new stance is the important thing, and so this whole section can be reduced to saying this.

On the issue of convergence between the digit proportions of plesiosaurs and Halszkaraptor, I will again state that any paper that suggests some sort of biomechanical similarity between a functional paddle and a classically coelurosaurian hand is simply incorrectly stating the anatomy. Regardless of what the morphometric analyses found with respect to relative lengths, there is no logically sound way of equating the morphology of the hand of Halszkaraptor with the manus of plesiosaurs and other marine reptiles. Nor should you draw comparisons with forelimb-propelled diving birds, which show extensive fusion and simplification in their forelimbs. Not only do the morphometric analyses in Cau et al. (2017) point to logically fallible comparisons, but they also (as stated in Brownstein (2019)) imply contrary locomotory modes, especially when the fact that Cau et al. (2017) seem to suggest H. escuilliei used its neck in undulatory locomotion is considered. The author may wish to include this section rebutting the rebuttal paper in further drafts of the manuscript, but I will continue to warn that this is simply poorly-performed comparative anatomy. That the author claims that the absence of a morphometric analysis in Brownstein (2019) testing those in Cau et al. (2017) implies the strength of the former is incorrect. Simply put, Halszkaraptor essentially has the same manual anatomy as any other coelurosaur.

On the issue of the ulna, the author has refused to conduct an additional morphometric analysis with the correction suggested, saying that mediolateral diameter has been considered a sufficient variable to test ulnar flattening in previous papers. Not only does the author not cite any particular papers, but this is simply incorrect geometrically. “Flattening” refers to the mediolateral width vs. the anteroposterior width. An ulna could be extremely robustly-built and have a short length and high mediolateral width value, and vice versa. What you need to test is how thin anteroposteriorly the ulna is relative to how wide it appears in dorsal and ventral views.

The author responded to my critique of the forelimb of Halszkaraptor being functionally equated to a paddle by saying that “There is not provided evidence that the complexity of the appendicular skeleton is relevant in shaping the morphologies discussed in this manuscript. In particular, note that many taxa with different phalangeal formulae and degrees of element fusion have converged independently to a flipper-like morphology.” The issue with this statement is that, in order for Halszkaraptor to be properly likened to these taxa, the major differences in anatomy must be considered.

On the point about pachystosis, the author claims “extreme” adaptations for a semiaquatic lifestyle were not claimed for Halszkaraptor, even though they refer to the features of the body plan they linked to such an ecology as aberrant novelities. Cau et al. (2017) claimed that Halszkaraptor was a piscivore with a highly modified rostrum that could use its neck and forelimb to locomote through the water and had a new walking stance on land. That is a highly derived ecology. In this way, the examination of possible pachystosis in Halszkaraptor using the data provided in the original study is important, especially since spinosaurids also developed this feature.

A significantly shortened first finger relative to the other two digits is found in a variety of paravians and basal avians. See Archaeopteryx. There is no evidence provided that this feature should not be expected in an early-diverging dromaeosaurid.

The digit II ungual of Halszkaraptor is clearly not as strongly recurved and developed as the same bone in derived dromaeosaurids, such as Deinonychus, Dakotaraptor, and Utahraptor. A simple comparison of the bone in Halszkaraptor with these shows the derived morphology is distinctive. The point about the non-log transformed analysis was not a recommendation to conduct one (which as the author notes should not be done), but instead to illustrate this issue. Some dromaeosaurids might have longer upper hindlimbs than others. What the author needs to test is the relative pedal proportions.

The three topological differences found in the rebuttal paper’s phylogeny were a sister taxon relationship between Halszkaraptorinae and Unenlagiinae and some polytomies of larger fasmilies early in maniraptoriform evolution. Contrary to the author, these several small differences are not “pivotal” to the results in the rebuttal paper, which aimed to show that the supposed adaptations for a semiaquatic ecology in Halszkaraptor were poorly interpreted and that some could be better explained as possible plesiomorphies of larger maniraptoran clades.
If the author is failing to replicate the results presented in the rebuttal, it must have to do with the matrix itself, as it was not changed from Cau et al. (2015).

On the subject of the optimization analyses, I will not repeat what I said in the previous review. Simply put, nearly half of the features considered are probably plesiomophies of larger coelurosaurian clades. Even if some of these were modified slightly in Halszkaraptor, there is not a reason to suggest they indicate a semiaquatic ecology in that taxon. The alternative scenario suggested in the rebuttal paper was one in which Halszkaraptor retained features plesiomorphic with respect to larger clades of coelurosaurian dinosaur. This analysis essentially supports that for several characters.

The author claims that the undulatory and forelimb-propelled swimming modes in Halszkaraptor do not conflict with each other. Without having gone through and carefully performed biomechanical analyses and reconstructions, without taking the time to assess whether these two locomotory modes found in dramatically different taxa would fit together in Halszkaraptor by means of quantitative study, it is unjustified for the author to claim any sort of conclusion on whether these modes could exist in this taxon together. There is not any quantitative evidence for the undulatory mode presented here or in the original paper.

On the issue of what “semiaquatic” means, I cite the commentary of the other reviewers, who also note that this term is used confusingly in Cau et al. (2017) and potentially misleadingly here.

On the size of the anterior maxillary teeth, this feature is also found in basal troodontids (e.g., Xu and Norell, 2004; Pei et al., 2017). In any case, the slightly smaller size of the anterior maxillary teeth should not be used to support any ecological change in a species of coelurosaur, as tooth counts and tooth size are so highly variable (even ontogenetically) that this evidence is dubious.

The author has added a section to the paper comparing Halszkaraptor to sawbills. This is all well and good, but is presented ad hoc as another example of convergence between the former and semiaquatic avians that is not supported by any comparative figures, quantitative analysis, etc.

Based on the points raised here and in my previous review, I am not ready to accept the arguments made here unless the author at least attempts to conduct some of the modifications to the quantitative analyses I propose. The severe methodological issues noted above must be addressed.

All the best,

Chase Brownstein
Research Associate
Stamford Museum and Nature Center

·

Basic reporting

No comment.

Experimental design

No comment.

Validity of the findings

No comment.

Additional comments

No comment.